# Fast Conditional Mixing of MCMC Algorithms for Non-log-concave Distributions

**Xiang Cheng**[*]
MIT
x.cheng@berkeley.edu

**Bohan Wang**[*]
USTC
bhwangfy@gmail.com

**Jingzhao Zhang**[*]
IIIS, Tsinghua; Shanghai Qizhi Institute
jingzhaoz@mail.tsinghua.edu.cn

**Yusong Zhu**[*]
Tsinghua University
zhuys19@mails.tsinghua.edu.cn

## Abstract

MCMC algorithms offer empirically efficient tools for sampling from a target distribution $\pi(x) \propto \exp(-V(x))$. However, on the theory side, MCMC algorithms suffer from slow mixing rate when $\pi(x)$ is non-log-concave. Our work examines this gap and shows that when Poincaré-style inequality holds on a subset $\mathcal{X}$ of the state space, the conditional distribution of MCMC iterates over $\mathcal{X}$ mixes fast to the true conditional distribution. This fast mixing guarantee can hold in cases when global mixing is provably slow. We formalize the statement and quantify the conditional mixing rate. We further show that conditional mixing can have interesting implications for sampling from mixtures of Gaussians, parameter estimation for Gaussian mixture models and Gibbs-sampling with well-connected local minima.

## 1 Introduction

Sampling from a given target distribution of the form $\pi(x) \propto e^{-V(x)}$ plays a central role in many machine learning problems, such as Bayesian inference, optimization, and generative modeling [9, 14, 15]. The Langevin MCMC algorithm in particular has received a lot of recent attention; it makes use of the first-order gradient information $\nabla V(x)$, and can be viewed as the sampling analog of gradient descent.

Langevin MCMC has been shown to converge quickly when $\pi(x)$ is log-concave [6, 10]. More recently, similar guarantees have been established for $p(x)$ satisfying weaker conditions such as log-Sobolev inequality (LSI) [23], for instance, $\pi(x) \propto e^{-V(x)}$ can have a good LSI constant when $V(x)$ is a small perturbation of a convex function. However, few guarantees exist for general non-log-concave distributions. One simple example is when $p(x)$ is a well-separated mixture of two Gaussian distributions; in this case, one verifies that Langevin MCMC mixes at a rate proportional to the inverse of the exponential of the separation distance.

Many important modern machine-learning problems are highly non-convex, one prominent class being functions arising from neural networks. Though finding the global minimum of a non-convex function can be difficult, gradient descent can often be shown to converge rather quickly to a *local optimum* [3]. This raises the important question:

> *What is the sampling analog of a local minimum? And how can we sample efficiently from such a minimum?*

---

[*]Alphabetical author order

37th Conference on Neural Information Processing Systems (NeurIPS 2023).

In [3], authors provide a partial answer to this question by adopting the view of Langevin Diffusion as the *gradient flow of KL divergence in probability space*. Under this perspective, the *gradient* of KL divergence is given by the Fisher Information (FI). Authors of [3] show that LMC achieves $\epsilon$ error in FI in $O(1/\epsilon^2)$ steps.

However, one crucial question remains unanswered: how does the local optimality of FI help us sample from non-convex distributions? Intuitively, small FI is useful for characterizing local convergence when $\pi$ is multi-modal. Authors of [3] suggested this connection, but it remains unclear how local mixing is defined and how *small FI* can quantitatively lead to *good local mixing*. Thus motivated, one of the goals of this paper is to provide a useful interpretation of the FI bound.

To this end, we propose a rigorous quantitative measure of "local mixing". We also provide a more general notion of "stationary point" for the sampling problem. Under these definitions, we show that LMC can achieve $\epsilon$ error in our proposed "measure of local mixing", in polynomial time. Finally, we consider discrete-time analog of the aforementioned ideas, and prove local convergence for random walk on a hypercube. Below is a detailed description of our theoretical contributions.

### 1.1 Main Contributions

1. We define a notion of *conditional convergence*: Let $\mathcal{X} \subseteq \Omega$ denote a subset of the state space. We study the convergence of $\pi_t | \mathcal{X}$ to $\pi | \mathcal{X}$, where $\pi_t$ denotes distributions of LMC iterates and $\pi$ denotes the target distribution. This definition of convergence is much weaker than the standard global convergence, but in exchange, LMC can achieve fast conditional convergence in settings where global convergence is known to be exponentially slow.

2. We define local Logarithmic Sobolev Inequality and show how to combine it with existing results on the convergence of Fisher information to derive the conditional convergence of LMC.

3. When local Logarithmic Sobolev Inequality does not hold, we define local Poincaré Inequality and Poincaré Fisher information (which is an analogy of Fisher information). We show the convergence of Poincaré Fisher information assuming strong dissipativity and show the conditional convergence of LMC when local Poincaré Inequality is present.

4. To showcase the applications of our results, we respectively study sampling from Gaussian mixture model with the same covariance and sampling from the power posterior distribution of symmetric two-component Gaussian mixtures. The global isometric constants of these examples are exponential in dimension and may have slow global convergence. We show that the local isoperimetric constants of these examples are polynomial in dimension, and prove fast conditional convergence for these examples.

5. In Theorem 1, we consider an application of our result to Gibbs sampling on discrete state space. We show that fast conditional mixing happens when the spectral gap is not small. We further show in Theorem 2 that a subset has large spectral gap if it contains only one local minimum and is well connected.

## 2 Related Work

When the target distribution $\pi$ is strongly log-concave, the entropy $\mathrm{Ent}_\pi\left[\frac{\mu}{\pi}\right]$ is known to be strongly convex with respect to $\mu$, and thus Langevin Dynamics converges exponentially fast [2]. Such a result is later extended to LMC [8, 4, 9, 7, 21]. Several works further loosen the assumption by using isoperimetric inequalities such as Logarithmic Sobolev Inequality and Poincaré Inequality instead of strong log-concavity [24, 5, 25, 11]. However, there are few existing works on general non-log-concave sampling. Recently, Balasubramanian [3] defines convergence in relative Fisher information as a kind of "weak convergence" for sampling and proves a polynomial guarantee in general non-log-concave case only assuming global Lipschitz condition. However, this paper doesn't give any rigorous statistical interpretation of this weak convergence; Majka et al. [18] and Erdogdu et al. [11] study the cases when the target distribution is non-log-concave but has some good tail growth or curvature; Ma et al. [17] analyze the situation when the target distribution is non-log-concave inside the region but log-concave outside. Although these works give strict proofs of polynomial time guarantee in their setting, their results only hold for a small branch of non-log-concave distributions. It is still hardly possible to obtain a polynomial guarantee in general

non-log-concave cases. Multimodality, as a special case of non-log-concavity, has attracted lots of attention due to its prevalence in applied science. Many modified versions of MCMC were proposed to try to tackle the sampling of these distributions, such as Darting MC [1], Wormhole HMC [16], and etc. However, these algorithms require explicit knowledge of the location of the modes.

## 3 Conditional Mixing for MCMC

In this section, we provide a formal definition of "conditional mixing". Specifically, let $\mu_t$ denote the distribution of a Markov chain $\{Z_t\}_t$ at time $t$. We assume that the Markov chain dynamics is reversible with a unique stationary distribution $\pi$. Existing analyses mostly focus on understanding the rate of convergence measured by $d(\mu_t, \pi)$ where $d$ is some probability distance.

However, unless the stationary distribution $\pi$ satisfies certain restrictive properties (e.g., log-concavity), the rate of convergence can be exponentially slow in the problem dimension or the distribution moments even for simple distributions such as the mixture of Gaussians. For this reason, we consider a weaker notion of convergence below.

**Definition 1** (Conditional mixing). *Given a distribution $\mu_t$ supported on the state space $\Omega$. We say $\mu_t$ converges conditioned on set $\mathcal{X} \subseteq \Omega$ with respect to the divergence $d$ if*

$$d(\mu_t | \mathcal{X}, \mu | \mathcal{X}) \leq \epsilon,$$

*where we have the conditional distribution*

$$\mu_t | \mathcal{X}(x) = \frac{\mu_t(x) \mathbb{1}\{x \in \mathcal{X}\}}{\mu_t(\mathcal{X})}.$$

For now we can think of the distance $d$ as the total variation distance. Later we will discuss stronger divergence such as KL-divergence or Chi-squared divergence.

The focus of our work is on identifying several sufficient conditions for fast convergence, and quantitatively bounding the convergence rate under these conditions. We focus on two MCMC algorithms: the Langevin Monte Carlo in continuous space, and the Gibbs sampling algorithm in discrete space. We further discuss the implications of conditional convergence for the two algorithms.

## 4 Conditional Mixing for Langevin Monte Carlo

In this section, we study the conditional mixing of the Langevin Monte Carlo (LMC) algorithm. This section is organized as follows: in Subsection 4.1, we first introduce the Langevin Monte Carlo algorithm, Langevin Dynamics, functional inequalities, and the Fisher information; in Subsection 4.2, we provide our main results characterizing the conditional convergence of LMC; finally, in Subsection 4.3, we showcase two applications of our main results.

### 4.1 Preliminaries

**Langevin Monte Carlo.** We are interested in the convergence of Langevin Monte Carlo (LMC), which is a standard algorithm employed to sample from a target probability density $\pi \propto e^{-V} : \mathbb{R}^d \to \mathbb{R}$, where $V$ is called the *potential function*. The pseudocode of LMC is given in Algorithm 1.

---
**Algorithm 1** Langevin Monte Carlo

---
**Input:** Initial parameter $z$, potential function $V$, step size $h$, number of iteration $T$

1: Initialization $z_0 \leftarrow z$
2: **For** $t = 0 \to T$:
3:     Generate Gaussian random vector $\xi_t \sim \mathcal{N}(0, \mathbb{I}_d)$
4:     Update $z_{(t+1)h} \leftarrow z_{th} - h\nabla V(z_{th}) + \sqrt{2h}\xi_t$
5: **EndFor**

---

LMC can be viewed as a time-discretization of Langevin Dynamics (LD), described by the following stochastic differential equation:

$$\mathrm{d}Z_t = -\nabla V(Z_t)\,\mathrm{d}t + \sqrt{2}\mathrm{d}B_t. \tag{1}$$

We can interpolate LMC following the similar manner of LD as

$$dZ_t = -\nabla V(Z_{kh})\,dt + \sqrt{2}\,dB_t,\ t \in [kh, kh+1),\ k \in \mathbb{N}. \tag{2}$$

One can easily observe that $\{Z_{kh}\}_{k=0}^{\infty}$ in Eq. (2) has the same joint distribution as $\{z_{kh}\}_{k=0}^{\infty}$ in Algorithm 1, and thus Eq. (2) is a continuous-time interpolation of Algorithm 1.

**Poincaré Inequality & Logarithmic Sobolev Inequalities.** The convergence of LMC does not necessarily hold for all potential functions, and quantitative bounds require specific conditions over the potential function $V$. Poincare Inequality (PI) and Logarithmic Sobolev Inequality (LSI) are two commonly used conditions in the analysis of LMC convergence. We present these below:

**Definition 2** (Poincaré Inequality). *A probability measure $\pi$ on $\mathbb{R}^d$ satisfies the Poincaré inequality with constant $\rho > 0$ (abbreviated as $\mathrm{PI}(\rho)$), if for all functions $f : \mathbb{R}^d \to \mathbb{R}$,*

$$\int \|\nabla f(x)\|^2\,d\pi(x) \geq \rho \mathrm{Var}_\pi[f]. \tag{PI}$$

**Definition 3** (Logarithmic Sobolev Inequality). *Given a function $f : \mathbb{R}^d \to \mathbb{R}^+$ and a probability measure $\pi$ on $\mathbb{R}^d$, define $\mathrm{Ent}_\pi(f) \triangleq \int f \log f\,d\pi(x) - \int f\,d\pi(x)\left(\log \int f\,d\pi(x)\right)$. We say that a distribution $\pi$ on $\mathbb{R}^d$ satisfies the Logarithmic Sobolev Inequality (abbreviated as $\mathrm{LSI}(\alpha)$) with some constant $\alpha$, if for all functions $f : \mathbb{R}^d \to \mathbb{R}^+$,*

$$\int_{\mathcal{X}} \frac{\|\nabla f(x)\|^2}{f}\,d\pi(x) \geq \alpha \,\mathrm{Ent}_\pi(f). \tag{LSI}$$

Both PI and LSI can imply the convergence of LMC when the step size $h$ is small. **Specifically, denote $\pi_{th}$ as the distribution of $z_{th}$ in LMC (Algorithm 1) and $\tilde{\pi}_t$ as the distribution of $Z_t$ in Langevin Dynamics (Eq.(1)).** We further denote $\pi_t$ as the distribution of $Z_t$ interpolating LMC. The left-hand-side of Eq.(PI) with $f = \frac{\tilde{\pi}_t}{\pi}$ is then the derivative (w.r.t. time) of the entropy of $f$, i.e., $\mathrm{Ent}_\pi\left[\frac{\tilde{\pi}_t}{\pi}\right]$, and thus we directly establish the exponential convergence of $\mathrm{Var}_p[\frac{\tilde{\pi}_t}{\pi}]$. The convergence of LMC can then be induced by bounding the discretization error between LMC and Langevin Dynamics. The methodology is similar when we have LSI.

**Fisher information.** It is well-known that if $V$ is either strongly convex or convex with bounded support, then it obeys both PI and LSI, and the convergence of LMC follows immediately according to the arguments above. However, real-world sampling problems usually have non-convex potential functions, and for these problems we may no longer have either PI or LSI. If we revisit the above methodology to establish the convergence of LMC under LSI, we find that we still have

$$\frac{d}{dt}\mathrm{Ent}_\pi\left[\frac{\tilde{\pi}_t}{\pi}\right] = -\int \left\|\nabla \ln \frac{\tilde{\pi}_t}{\pi}(x)\right\|^2 d\pi_t(x), \mathrm{Ent}_\pi\left[\frac{\tilde{\pi}_0}{\pi}\right] - \mathrm{Ent}_\pi\left[\frac{\tilde{\pi}_T}{\pi}\right] = \int_0^T \int \left\|\nabla \ln \frac{\tilde{\pi}_t}{\pi}(x)\right\|^2 d\pi_t(x)\,dt,$$

and thus $\lim_{T\to\infty} \min_{t\in[0,T]} \int \left\|\nabla \frac{\tilde{\pi}_t}{\pi}(x)\right\|^2 d\pi(x) = 0$. Following this methodology, [3] uses the considers a notion of convergence which is defined using Fisher Information $\mathrm{FI}(\mu\|\pi) \triangleq \int \|\nabla \ln \mu/\pi\|^2\,d\mu$, which they use to analyze LMC convergence. We present the result of [3] for completeness:

**Proposition 1** (Theorem 2, [3]). *Assume $\nabla V$ is $L$-lipschitz. Then, for any step size $h \in (0, \frac{1}{6L})$,*

$$\frac{1}{Th}\int_0^{Th} \mathrm{FI}_\pi\left(\pi_t\|\pi\right)dt \leq \frac{2\,\mathrm{Ent}\left(\frac{\pi_0}{\pi}\right)}{Th} + 8L^2 dh.$$

## 4.2 Rates of convergence

### 4.2.1 Conditional mixing under local Logarithmic Sobolev Inequality

We first show that if the target distribution $\pi$ obeys a local Logarithmic Sobolev Inequality (defined below), then convergence of Fisher information implies conditional convergence.

**Definition 4** (Local Logarithmic Sobolev Inequality). *We say that a distribution $\pi$ on $\mathbb{R}^d$ satisfies the local Logarithmic Sobolev Inequality over a subset $S \subset \mathbb{R}^d$ with some constant $\alpha$ (abbreviated as $\mathrm{LSI}_S(\alpha)$), if $\pi|S$ satisfies $LSI(\alpha)$.*

Definition 4 characterizes the local property of one distribution. It is considerably weaker than global LSI (i.e., Definition 3) when $S$ is convex, and recovers LSI when $S = \mathbb{R}^d$. The intuition is that when the distribution is multi-modal, it is quite possible that LSI does not hold (or hold but with an exponentially small constant). In contrast, we can reasonably expect local LSI hold within each mode. In Subsection 4.3, we will show that a Gaussian mixture model with the same covariance satisfies Local Logarithmic Sobolev Inequality. We show in the following lemma that, whenever we have an estimation on the Fisher information between $\mu$ and $\pi$, we can turn it to an estimation of the entropy between $\mu|S$ and $\pi|S$ given $\text{LSI}_S(\alpha)$.

**Lemma 1.** *Let $S \subset \mathcal{X}$ and assume that $\pi$ obeys $\text{LSI}_S(\alpha)$. For any distribution $\mu$ such that $\text{FI}(\mu||\pi) \leq \varepsilon$, we have that either $\mu(S) \leq \frac{\sqrt{\varepsilon}}{\sqrt{\alpha}}$, or $\text{Ent}_{\pi|S}[\frac{\mu|S}{\pi|S}] \leq \frac{\sqrt{\varepsilon}}{\sqrt{\alpha}}$.*

We defer the proof of Lemma 1 to Appendix A.1. Intuitively, Lemma 1 states that if the distribution $\pi$ satisfies local LSI over the region $S$ and has small global Fisher information with respect to a distribution $\mu$, one can expect that either the probability mass of $\mu$ over $S$ is minor, or $\mu$ is close to $\pi$ both conditional over $S$. In other words, $\mu$ is close to $\pi$ conditional over $S$ whenever $\mu$ has a moderate mass over $S$. Note that Proposition 1 has already guaranteed global FI of LMC with little assumption, which together with Lemma 1 leads to the following corollary:

**Corollary 1.** *Assume $\nabla V$ is $L$-lipschitz and $S \subset \mathcal{X}$ satisfies that $\pi$ obeys $\text{LSI}_S(\alpha)$. Define $\bar{\pi}_{Th} = \frac{\int_0^{Th} \pi_t \, dt}{Th}$. Choosing step size $h = \frac{1}{\sqrt{T}}$, we have that either $\bar{\pi}_{Th}(S) \leq \mathcal{O}\left( \frac{\sqrt{d} \sqrt[4]{\text{Ent}\left(\frac{\pi_0}{\pi}\right)}}{\sqrt{\alpha} \sqrt[4]{T}} \right)$, or*

$$\text{Ent}_{\pi|S} \left[ \frac{\bar{\pi}_{Th}|S}{\pi|S} \right] \leq \mathcal{O}\left( \frac{\sqrt{d} \sqrt[4]{\text{Ent}\left(\frac{\pi_0}{\pi}\right)}}{\sqrt{\alpha} \sqrt[4]{T}} \right).$$

Corollary 1 is one of our key results. Intuitively, it shows that LMC can achieve local mixing when only local LSI is assumed. As we discussed under Definition 4, we can reasonably expect that local LSI holds within each mode when the distribution is multi-modal, in which case Corollary 1 can be further applied to show local mixing with polynomial rate. Note here the dependence of the local mixing rate is polynomial over the dimension $d$, meaning that considering local mixing helps to get rid of the exponential dependence over the dimension $d$ in the global mixing rate when studying multi-modal distributions.

### 4.2.2   Conditional Convergence under local Poincaré Inequality

We have established the conditional convergence of LMC under local Logarithmic Sobolev Inequality above. However, there are cases where even local LSI fails to hold. To tackle these cases and inspired by Poincaré Inequality, we introduce a weaker notion than local LSI called local Poincaré Inequality:

**Definition 5** (Local Poincaré Inequality). *A distribution $\pi$ on $\mathbb{R}^d$ satisfies the local Poincaré Inequality over a subset $S \subset \mathbb{R}^d$ with constant $\rho$ (abbreviated as $\text{PI}_S(\rho)$), if $\pi|S$ satisfies $\text{PI}(\rho)$.*

As Poincaré Inequality is weaker than Logarithmic Sobolev Inequality, one can easily see that local Poincaré Inequality is also weaker than local Logarithmic Sobolev Inequality. Based on local Poincaré Inequality, we have the following Poincaré version of Lemma 1, which converts an estimation of Poincaré Fisher information to an estimation of chi-squared divergence of the conditional distributions.

**Lemma 2.** *Define Poincaré Fisher information as $\text{PFI}(\mu||\pi) \triangleq \int \|\nabla(\mu/\pi)\|^2 \, d\mu$. Let $S \subset \mathcal{X}$ and assume that $\pi$ obeys $\text{PI}_S(\rho)$. For any distribution $\mu$ such that $\text{PFI}(\mu||\pi) \leq \varepsilon$, we have that either $\mu(S)^2 \text{Var}_{\pi|S}[\frac{\mu|S}{\pi|S}] \leq \frac{\varepsilon\pi(S)}{\rho}$.*

To derive the conditional convergence of LMC under local Poincaré inequality, we further need to bound Poincaré Fisher information of LMC. Such a result, however, does not exist in the literature to our best knowledge. In analogy to the analysis in [3], we develop the following lemma to control the Fisher information of LD (recall that $\tilde{\pi}_t$ is the distribution of Langevin Dynamics in time $t$).

**Proposition 2.** *Denote $\bar{\tilde{\pi}}_{Th} = \frac{\int_0^{Th} \tilde{\pi}_t \, dt}{Th}$. Then, we have the following estimation of the PFI between $\bar{\tilde{\pi}}_{Th}$ and $\pi$.*

$$\text{PFI}(\bar{\tilde{\pi}}_{Th}||\pi) \, dt \leq \frac{\text{Var}_\pi[\frac{\tilde{\pi}_0}{\pi}]}{2Th}.$$

The proof can be directly derived by taking the derivative of $\mathrm{Var}_\pi[\frac{\tilde{\pi}_t}{\pi}]$ with respect to $t$. We defer the formal proof to Appendix A.2. Combining Lemma 2, Proposition 2, and recent advances in discrete error analysis of LD [19], we obtain the following result showing conditional convergence under local Poincaré inequality.

**Corollary 2.** *Assume $\nabla V$ and $\nabla^2 V$ is $L$-lipschtiz, $V$ satisfies strong dissipativity, i.e., $\langle \nabla V(x), x \rangle \geq m\|x\|^2 - b$, and $S \subset \mathcal{X}$ satisfies that $\pi|_S$ obeys $\mathrm{PI}(\alpha)$. Initialize $z_0 \sim \mathcal{N}(0, \sigma^2\mathbb{I})$ and select $h = \tilde{\Theta}\left( \frac{\mathrm{Var}_\pi[\frac{\tilde{\pi}_0}{\pi}]^{\frac{1}{3}}}{T^{\frac{2}{3}} \rho^{\frac{1}{3}} d^{\frac{2}{3}}} \right)$. Recall that $\bar{\pi}_{Th} = \frac{\int_0^{Th} \pi_t \, dt}{Th}$. Then either $\bar{\pi}_{Th}(S) = \mathcal{O}(\frac{d^{\frac{1}{6}} \mathrm{Var}_\pi[\frac{\tilde{\pi}_0}{\pi}]^{\frac{1}{6}}}{\rho^{\frac{1}{6}} T^{\frac{1}{12}}})$, or*
$$D_{TV}\left[\bar{\pi}_{Th}|S\|\pi|S\right] = \mathcal{O}(\frac{d^{\frac{1}{6}} \mathrm{Var}_\pi[\frac{\tilde{\pi}_0}{\pi}]^{\frac{1}{6}}}{\rho^{\frac{1}{6}} T^{\frac{1}{12}}}).$$

Corollary 2 shows that conditional mixing can be established when local PI holds, which is a weaker assumption than local LSI. However, the assumption of Corollary 2 is stronger than its LSI counterpart, and the mixing rate depends on $\mathrm{Var}_\pi[\frac{\tilde{\pi}_0}{\pi}]$ instead of $\mathrm{Ent}\left(\frac{\tilde{\pi}_0}{\pi}\right)$, which can be considerably large. These drawbacks are mainly due to that the dynamics of chi-square distance are much harder than that of KL divergence, even when global PI holds [12].

**Extension to Rényi divergence.** In some prior works, e.g. [5, 23], Rényi divergence is considered a natural "measure" of convergence, due to the analytical simplicity of the resulting expressions. One may wonder if our methodology above can be extended to establish a local mixing rate of Rényi divergence. Unfortunately, there are technical difficulties which for now we have not find a way to tackle or bypass, which we list in Appendix A.3.

## 4.3 Applications

In this subsection, we apply Corollary 1 and Corollary 2 to two concrete examples to demonstrate their usage in obtaining local mixing rates. We start by analyzing the example of sampling from Gaussian Mixture Models with uniform covariance and then turn to the example of sampling from the power posterior distribution of symmetric two-component Gaussian mixtures.

### 4.3.1 Case Study: Sampling from Gaussian Mixture Model with the uniform covariance

**Target distribution and potential function.** We are interested in the gaussian mixture model formed by gaussians with the same variance but different means. Specifically, the target distribution is defined as $\pi = \sum_{i=1}^n w_i p_i \in \Delta(\mathbb{R}^d)$, where $w_i > 0$, $\sum_{i=1}^n w_i = 1$, $p_i \sim \mathcal{N}(\mu_i, \Sigma)$ and $\Sigma \succ \sigma^2 \mathbb{I}_d$. The potential function is then defined as $V(x) = -\log(\sum_{i=1}^n w_i p_i(x))$.

**Partition of Space.** We divide the space according to the sub-level set. Specifically, we define $S_i \triangleq \{x : p_i(x) \geq p_j(x), \forall j \neq i\}$. One can easily verify that $\cup_{i=1}^n S_i = \mathbb{R}^d$. Furthermore, $S_i$ is convex since by the definition of $p_i$, we have

$$S_i = \{x : (x - \mu_i)^\top \Sigma^{-1}(x - \mu_i) \leq (x - \mu_j)^\top \Sigma^{-1}(x - \mu_j), \forall j \neq i\}.$$

As $\Sigma$ is positive definite and symmetric, we can decompose $\Sigma$ into $UU$, where $U$ is also positive definite and symmetric. We then obtain

$$U^{-1}S_i = \{x : (x - U^{-1}\mu_i)^\top(x - U^{-1}\mu_i) \leq (x - U^{-1}\mu_j)^\top(x - U^{-1}\mu_j), \forall j \neq i\},$$

and thus $U^{-1}S_i$ is one region of the Voronoi diagrams generated by $\{U^{-1}\mu_i\}_{i=1}^n$, which is convex. As $S_i$ can be obtained by performing linear transformation to $U^{-1}S_i$, we obtain that $S_i$ is also convex.

**Verification of local Logarithmic Sobolev Inequality.** We prove the local Logarithmic Sobolev Inequality of $\pi$ over each partition $S_i$ as follows.

**Lemma 3.** *For all $i \in \{1, \cdots, n\}$, $\pi|S_i$ obeys $\mathrm{LSI}(\frac{\sigma^{-2} \min_{i \in [n]} w_i}{\max_{i \in [n]} w_i})$.*

Lemma 3 is proved by first showing $p_i|S_i$ obeys $\mathrm{LSI}(\sigma^{-2})$ through Bakry-Émery criterion due to the convexity of $S_i$ and $p_i$ is strongly convex, and then applying Holley-Stroock perturbation principle by viewing $\pi$ as a perturbation of $p_i$ over $S_i$. The concrete proof is deferred to Appendix B.1.

**Verification of Lipschitz gradient.** By direct calculation, we derive the following bound on the Lipschitz constant of $\nabla V$.

**Lemma 4.** $\forall x \in \mathbb{R}^d$, $\|\nabla^2 V(x)\| \leq \frac{\max_{i,j} \|\mu_i - \mu_j\|^2}{\sigma^4} + \sigma^{-2}$.

As a conclusion, we obtain the following guarantee of running LMC over this example.

**Corollary 3.** *Let the stepsize of LMC be $h = \Theta(\frac{1}{\sqrt{T}})$. Assume $T > \Theta(\frac{d^2}{\varepsilon^4} \frac{\max_{i \in [n]} w_i^2}{\sigma^{-4} \min_{i \in [n]} w_i^2})$. Then, for every $i \in [n]$, either $\bar{\pi}_{Th}(S_i) \leq \varepsilon$, or $\mathrm{Ent}_{\pi|S_i}\left[\frac{\bar{\pi}_{Th}|S_i}{\pi|S_i}\right] \leq \varepsilon$.*

Corollary 3 indicates that running LMC over gaussian mixture distribution (with the same covariance) can ensure local mixing with cost polynomial over the dimension and independent of the distance between the means. By contrast, it is a folk-tale that even the global mixing of gaussian mixture with 2 components has a global mixing rate exponential over the distance, which suggests local mixing analysis provide a more concrete analysis of what is going on over each component.

#### 4.3.2 Sampling from the power posterior distribution of symmetric Gaussian mixtures

**Target distribution and potential function.** The symmetric Gaussian mixture model is given as

$$f_{\theta_0}(x) \triangleq \frac{1}{2}\varphi(x; \theta_0, \mathbb{I}_d) + \frac{1}{2}\varphi(x; -\theta_0, \mathbb{I}_d).$$

Here $\varphi(x; \theta_0, \mathbb{I}_d)$ denotes the multivariate Gaussian distribution with location parameter $\theta_0 \in \mathbb{R}^d$ and covariance matrix $\mathbb{I}_d$. Without loss of generality, we assume $\theta_0 \in \mathrm{span}\{e_1\}$, i.e., all coordinates of $\theta_0$ except the first is zero, and $\theta_0 = \|\theta_0\| e_1$. The power posterior distribution, or the target distribution, is defined as $\pi_{n,\beta/n}\left(\theta \mid \{X_i\}_{i=1}^n\right) := \frac{\prod_{i=1}^n (f_\theta(X_i))^{\beta/n} \lambda(\theta)}{\int \prod_{i=1}^n (f_u(X_i))^{\beta/n} \lambda(u) du}$. We set $\lambda \equiv 1$ for simplicity. When no ambiguity is possible, we abbreviate $\pi_{n,\beta/n}\left(\theta \mid \{X_i\}_{i=1}^n\right)$ as $\pi(\theta)$. The potential function is then given as $V(\theta) := \frac{\beta}{n}\sum_{i=1}^n \log\left(\frac{1}{2}\varphi(\theta - X_i) + \frac{1}{2}\varphi(\theta + X_i)\right) + \log\lambda(\theta)$.

**Partition of Space.** With an accuracy budget $\varepsilon$, we define $R_1 = [0, A] \times \mathcal{B}(0, M)$, $R_2 = [-A, 0] \times \mathcal{B}(0, M)$, and $R_3 = (R_1 \cup R_2)^c$, where $A$ and $M$ are $\varepsilon$-dependent hyperparameter specified latter.

**Verification of local Poincaré Inequality over $R_1$ and $R_2$.** We have the following characterization for the local Poincaré Inequality over $R_1$ and $R_2$:

**Lemma 5.** *If $A, M \geq \|\theta_0\| + 1$ and $n \geq \tilde{\Theta}((A + M)^2 d \log(1/\delta))$, then with probability at least $1 - \delta$ with respect to the sampling of $\{X_i\}_{i=1}^n$, we have that $\pi$ obeys $\mathrm{PI}_{R_1}(\Theta(1/(A^4 M^2)))$ and $\mathrm{PI}_{R_2}(\Theta(1/(A^4 M^2)))$.*

The lemma is derived by first considering the distribution $\bar{\pi}$ corresponding to the potential function $\bar{V} = \mathbb{E}V$ and proving local Poincaré Inequality of $\bar{\pi}|R_1$ (or $\bar{\pi}|R_2$), then bounding the difference between $\bar{V}$ and $V$ through concentration inequality, and finally completing the proof by applying Holley-Stroock perturbation principle to pass local Poincaré Inequality from $\bar{\pi}|R_1$ to $\pi|R_1$. A concrete proof is deferred to Appendix B.2.

**Verification of strong dissipativity.** Similar to local Logarithmic Sobolev Inequality, we can show strong dissipativity of $V$ holds in high probability.

**Lemma 6.** *If $n > \tilde{\Theta}((d + \|\theta_0\|^2)\log(1/\delta))$, then with probability at least $1 - \delta$ over the sampling of $\{X_i\}_{i=1}^n$, $\|\nabla^2 V(\theta)\| \leq \mathcal{O}(1 + \|\theta_0\|^2)$, $\langle \nabla V(\theta), \theta \rangle \geq \Omega(\|\theta\|^2) - \mathcal{O}\left(\|\theta_0\|^2 + 1\right)$, and $\|\nabla^3 V(\theta)\| \leq \mathcal{O}(d)$.*

**Bounding the probability of $R_3$.** Using strong dissipativity, we can bound $\pi_t(R_3)$ as follows.

**Lemma 7.** *If $n > \tilde{\Theta}((d + \|\theta_0\|^2)\log(1/\delta))$, then with probability at least $1 - \delta$ over the sampling of $\{X_i\}_{i=1}^n$, we have $\pi_t(R_3) \leq 32 e^{\frac{16\beta\left(\|\theta_0\|^2 + 1\right) + 6d - \min\{A, M\}^2}{2\beta}}$.*

All in all, we obtain the following characterization of running LMC over this example.

**Corollary 4.** *Initialize $z_0 \sim \mathcal{N}(0, \sigma^2\mathbb{I})$ for $\sigma^2 \leq \frac{1}{1+L}$ and select $h = \tilde{\Theta}\left(\frac{1}{T^{\frac{2}{3}}}\right)$. Set $A = M = \Theta(d + \log(1/\varepsilon))$. If $T = \Omega(d^2/\varepsilon^{12})$ and $n > \tilde{\Theta}((d + \|\theta_0\|^2)\log(1/\delta))$, then with probability at least $1 - \delta$ over the sampling of $\{X_i\}_{i=1}^n$, either $\bar{\pi}_{Th}(R_i) \leq \varepsilon$ or $D_{TV}\left[\bar{\pi}_{Th}|R_i \| \pi|R_i\right] \leq \varepsilon$ for $i \in \{1, 2\}$, and $\bar{\pi}_{Th}(R_3) \leq \varepsilon$.*

As a comparison, such a problem was also studied by [20], where they constructed a specific sampling algorithm with prior knowledge of this problem to achieve global convergence. In contrast, we analyze LMC, which does not require any prior knowledge of the problem, and derive the conditional convergence of LMC.

## 5 Conditional Mixing for Gibbs Sampling on Finite States

In previous sections, we showed that conditional mixing can happen for LMC on a continuous state space. We now show that similar results hold for MCMC algorithms on a finite state space. For simplicity, we consider an energy function $f : \{0, 1\}^d \to \{0, 1, 2, ..., M\} =: [M]$ defined on the vertices of a hypercube. Denote its corresponding Gibbs measure $\pi(x) \propto e^{-f(x)}$.

We consider vertices of the hypercube as a d-regular graph where for any $x, y \in \{0, 1\}^d$, an edge exists $x \sim y$ if and only if they differ by one coordinate, $d_{\text{Hamming}}(x, y) = 1$. Then a *lazy Gibbs sampler* has the following transition matrix on this finite graph:

$$
p(x, y) = \begin{cases} \frac{1}{2d} \frac{\pi(y)}{\pi(y)+\pi(x)}, & y \sim x, \\ 1 - \frac{1}{2d} \sum_{x', \text{s.t. } x' \sim x} \frac{\pi(x')}{\pi(x')+\pi(x)}, & y = x, \\ 0, & \text{otherwise.} \end{cases}
$$

Note that the process is lazy because, $p(x, x) \geq 1/2$ for any $x$. This is assumed for simplicity of analysis to avoid almost periodic behaviors. This assumption does not make the analysis less general, as a lazy self loop with probability $1/2$ only changes the mixing time by a multiplicative absolute constant (see Corollary 9.5 [22]).

To prove conditional convergence, we need an analogue of conditional Poincaré Inequality. The story for the discrete state space can be more convoluted as the transition matrix, in addition to the stationary distribution, plays a role here. Many of the analyses below are inspired by [13].

First, given a finite number of subsets $\{\mathcal{X}_i\}_{i \leq m}$, we define the conditional probability $\pi_i = \pi | \mathcal{X}_i$ supported on $\mathcal{X}_i$, and hence for $w_i = \pi(\mathcal{X}_i)$,

$$
\pi(x) = \sum_{i=1}^{m} w_i \pi_i(x) \mathbb{1} \{x \in \mathcal{X}_i\}.
$$

We also need to design a conditioned transition kernel so that $\forall x, y \in \mathcal{X}_i$,

$$
p_i(x, y) = p(x, y) + \mathbb{1} \{x = y\} P(x, \mathcal{X}_i^c), \tag{3}
$$

where $\mathcal{X}_i^c$ denotes the complement of $\mathcal{X}_i$, and hence the conditioned kernel simply rejects all outgoing transitions. Then we can easily tell that $p_i$ is reversible with a unique stationary distribution $\pi_i$. We are now ready to give an analogue of Corollary 1.

**Theorem 1.** *Given a sequence of subsets $\{\mathcal{X}_i\}_{i \leq m}$. If for every $i$, $P_i$ defined in (3) as a distribution on $\mathcal{X}_i$ has spectral gap at least $\alpha$, then we have that either for some $t$, the distribution $\mu_t$ has small probability on $\mathcal{X}_i$, $\mu_t(\mathcal{X}_i) \leq \pi(\mathcal{X}_i)T^{-1/4}$, or the conditional mixing happens with respect to Chi-squared divergence,*

$$
\frac{1}{T} \sum_t \text{Var}_{\pi_i} \left[ \frac{\mu_t | \mathcal{X}_i}{\pi | \mathcal{X}_i} \right] \leq \frac{1}{\alpha \sqrt{T}} \text{Var}_{\pi} \left[ \frac{\mu_0}{\pi} \right].
$$

The proof builds upon the convergence of the Dirichlet form and can be found in Appendix C. The above theorem suggests that if the spectral gap of the Gibbs sampling algorithm is lower bounded on a local subset, then after a polynomial number of iterations, we get either $\mu_t$ has very small probability on this set, or conditioned on the set, the distribution is close to the stationary distribution.

One thing less clear, in finite-state Gibbs sampling as compared to the LMC, is when would the spectral gap for a Gibbs distribution be large. For the Langevin Monte Carlo algorithm, classical results show that the spectral gap cannot be too small if the stationary distribution is locally near log-concave. Below we provide an analogue of this observation for discrete state space.

## 5.1 Spectral Gap for Gibbs Sampling

We first define a graph $\mathcal{G} = (V, E)$, where $V = \{0, 1\}^d$, and $(x, y) \in E$ if and only if $d_{\text{Hamming}}(x, y) = 1$. Then we define quasi-concavity in this case. Note that Gibbs sampling in each iteration can only move to a neighbor or stay at the current vertex. Then we introduce the counterpart of quasi-convexity on a function defined on vertices of graphs.

**Definition 6.** *Let $\mathcal{G} = (V', E')$ be a subgraph, $V' \subseteq V$, $E' = (x, y) \in E | x \in V', y \in V'$. We say a function $f : V \to [M]$ is **quasi-convex with a radius D** on a subgraph $\mathcal{G} = (V', E')$, if there exists a local minimum $v* \in V'$ such that for any $v \in V'$, any shortest path $v \to v_1 \to ... \to v^*$ from $v \to v^*$ is of length at most $D$, and $f$ is non-increasing along the path.*

Before providing a guarantee for the spectral gap, we give two examples of quasi-convexity.

**Example 5.1.** *If $g : \mathbb{R}^+ \to \mathbb{R}$ is a quasi-convex function defined on positive reals. Then for a given $x^* \in V' \subseteq V$ any function $f(x) = g(a \cdot d(x, x^*) + b)$ is a quasi-convex function on the graph where $a, b$ are reals and $d(x, y)$ is the shortest path length from $x$ to $y$.*

**Example 5.2.** *If on a subset $V' \subseteq V$, $f(x) = c^T x$ for some $c \in \mathbb{R}^d, x \in \{0, 1\}^d$ (i.e. $f$ is a linear function), then $f$ is quasi-convex on the graph.*

The next theorem states that for such a function, the spectral gap is polynomial in problem dimension $d$ and the diameter of the region $D$.

**Theorem 2.** *If a function $f$ is quasi-convex with a radius $D$ on a subset $\mathcal{X}_i \subseteq \{0, 1\}^d$, then the conditional transition of Gibbs sampling defined in* (3) *has a spectral gap lower bounded by $\frac{1}{16d^2 D^2}$.*

The proof studies the conductance of the Markov chain and can be found in Appendix D. The above theorem suggests that although Gibbs sampling can mix very slowly, on any subset with a well connected local minimum, the conditional mixing to the stationary distribution can be fast. This concludes our analysis and we move on to verify our statements with experiments in the next section.

# 6 Experiments

## 6.1 Observations on Gaussian Mixture

In this section, we conduct experiments to verify the theoretical results and compare global mixing versus conditional mixing for Gaussian mixture models. We take three Gaussian mixtures: $\nu_1 = 0.9\mathbb{N}_1(-10, 1) + 0.1\mathbb{N}_1(10, 1)$, $\nu_2 = 0.15\mathbb{N}_1(-5, 1) + 0.15\mathbb{N}_1(-2.5, 1) + 0.3\mathbb{N}_1(0, 1) + 0.2\mathbb{N}_1(2.5, 1) + 0.2\mathbb{N}_1(5, 1)$, and $\nu_3 = 0.4\mathbb{N}_2((-5, -5), I_2) + 0.4\mathbb{N}_2((5, 5), I_2) + 0.1\mathbb{N}_2((-5, 5), I_2) + 0.1\mathbb{N}_2((5, -5), I_2)$ as our target distributions. We use Algorithm 1 as our sampling algorithm, and set step size $h = 10^{-2}$. The initial distributions are both uniform in a large enough range. We plot the sampling distribution after $T = 500, 5000, 500$ rounds respectively in Figure 1a, 1b, and 1c, and plot the conditional and global KL divergence in Figure 1d, 1e, and 1f.

We make the following observations: **(1)** The global KL divergences of the sampling distributions of $\nu_1$ and $\nu_3$ decrease fast at first. Then they maintain at a constant level and never converge to 0. It is reasonable since $\nu_1, \nu_3$ have very bad LSI constants (exponential in the distance between means). Thus, by classic Langevin Danymics analysis results[23], global KL divergence would have an exponentially slow convergence rate. **(2)** Both the global and conditional divergences of the sampling distribution of $\nu_2$ converge very fast. This is because dense Gaussian mixtures like $\nu_2$ have good LSI constant. The KL values are noisy due to the limited calculation precision of KL divergence. **(3)** The conditional divergence of the sampling distribution of $\nu_2$ converges faster than the global divergence because the LSI constant bound of $\nu_2$ is much worse than the conclusion in Corollary 1, which means conditional convergence is faster than global convergence. **(4)** The conditional KL divergences of the sampling distributions of $\nu_1$ and $\nu_3$ converge to 0 very fast (the flat part is due to the limit of calculation precision), which could be seen as a verification of our theoretical result. **(5)** The sampling distributions of $\nu_1$ and $\nu_3$ after $T$ iterations contain all of the Gaussian components in target distributions, the only difference between them is weight. Since learning the right weight and component will directly lead to global KL convergence, this observation could be seen as an example of the gap between global convergence and conditional convergence.

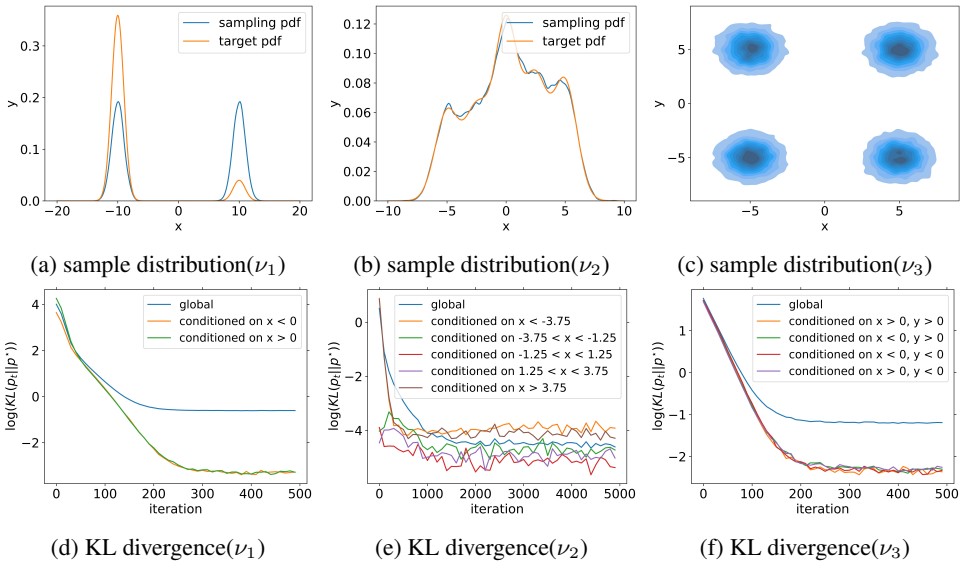

(a) sample distribution($\nu_1$)     (b) sample distribution($\nu_2$)     (c) sample distribution($\nu_3$)

(d) KL divergence($\nu_1$)     (e) KL divergence($\nu_2$)     (f) KL divergence($\nu_3$)

Figure 1: we plot the sampling distributions after $T$ iterations and the KL divergences w.r.t $t$

## 6.2 Observations on LMC with restarts

In the LMC analysis, We study the evolution of a distribution $p_0$ that usually is an absolutely continuous distribution. However, in practice, a more common implementation is as below: one first randomly generates an initial point $x_0$, runs the sampling algorithm for $T$ iterations, and then collects samples along a single trajectory. A gap between theory and practice here is that we always assume continuity and smoothness conditions on the initial distribution in theory, while in practice, we only generate a limited number of initial points (sometimes only one point) to run the sampling algorithm.

For log-concave sampling, this gap is usually negligible since the ergodicity of Langevin Dynamics guarantees that we could always capture the features of the target distribution. Thus, it's reasonable that there are plenty of works about the discretization error on the time scale, while we hardly pay attention to the approximation error of initial distribution. When it comes to non-log-concave sampling, this gap may become crucial. We conduct several experiments in Appendix E to verify this conjecture and show that LMC with restarts could empirically help to eliminate the gap and improve the convergence speed.

## 7 Conclusions

Our work examines sampling problems where the global mixing of an MCMC algorithm is slow. We show that in such cases, fast conditional mixing can be achieved on subsets where the target distribution has benign local structures. We make the above statements rigorous and provide polynomial-time guarantees for conditional mixing. We give several examples, such as the mixture of Gaussian and the power posterior to show that the benign local structure often exists despite the global mixing rate is exponentially slow.

Much remains to be done. Theoretically, whether faster convergence rates can be achieved or a lower bound exists remain unknown. Instantiating our analyses to more MCMC algorithms may also lead to new observations. More importantly, the implication of being able to sample efficiently from local distributions requires more careful analysis. This may lead to a new theoretical guarantee for problems with symmetry (such as permutation symmetry, sign symmetry, rotation invariance, etc) where all local minima are equally good and sampling from any mode suffices.

## Acknowledgments and Disclosure of Funding

Jingzhao Zhang acknowledges support from Tsinghua University Initiative Scientific Research Program.

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

# A Proofs of results in Section 4.2

## A.1 Local LSI: Proof of Lemma 1

*Proof.* Since

$$\mu(S) \int_S \left\| \nabla \frac{\mu|_S}{\pi|_S}(x) \right\|^2 \frac{\pi|_{\mathcal{X}_j}(x)^2}{\mu|_{\mathcal{X}_j}(x)} \, \mathrm{d}x = \int_S \left\| \nabla \frac{\mu}{\pi}(x) \right\|^2 \frac{\pi(x)^2}{\mu(x)} \, \mathrm{d}x \le \int_{\mathcal{X}} \left\| \nabla \frac{\mu}{\pi}(x) \right\|^2 \frac{\pi(x)^2}{\mu(x)} \, \mathrm{d}x,$$

if $\mu(S) \le \sqrt{\frac{\varepsilon}{\alpha}}$, the proof is finished. Otherwise, we have

$$\alpha \operatorname{Ent}_{\pi|_S} \left[ \frac{\mu|_S}{\pi|_S} \right] \le \int_S \left\| \nabla \frac{\mu|_S}{\pi|_S}(x) \right\|^2 \frac{\pi|_{\mathcal{X}_j}(x)^2}{\mu|_{\mathcal{X}_j}(x)} \, \mathrm{d}x \le \sqrt{\varepsilon\alpha}.$$

The proof is completed. $\square$

## A.2 Local PI: Proof of Proposition 2

To begin with, we provide the proof of Lemma 2.

*Proof of Lemma 2.* Since

$$\frac{\mu(S)^2}{\pi(S)} \int_S \left\| \nabla \frac{\mu|_S}{\pi|_S}(x) \right\|^2 \pi|_S(x) \, \mathrm{d}x = \int_S \left\| \nabla \frac{\mu}{\pi}(x) \right\|^2 \pi(x) \, \mathrm{d}x \le \int_{\mathcal{X}} \left\| \nabla \frac{\mu}{\pi}(x) \right\|^2 \pi(x) \, \mathrm{d}x.$$

The proof is completed by noticing that

$$\int_S \left\| \nabla \frac{\mu|_S}{\pi|_S}(x) \right\|^2 \pi|_S(x) \, \mathrm{d}x = \operatorname{PFI}(\mu|_S \| \pi|_S) \ge \rho \operatorname{Var}_{\pi|_S} \left[ \frac{\mu|_S}{\pi|_S} \right].$$

$\square$

We then prove Proposition 2.

*Proof of Propsition 2.* By applying the definition of LD, we have

$$\frac{\mathrm{d}\operatorname{Var}_\pi[\frac{\tilde{\pi}_t}{\pi}]}{\mathrm{d}t} = -2 \operatorname{PFI}(\tilde{\pi}_t \| \pi).$$

Taking integration of both sides of the above equation then gives

$$\operatorname{Var}_\pi \left[ \frac{\tilde{\pi}_0}{\pi} \right] \ge \operatorname{Var}_\pi \left[ \frac{\tilde{\pi}_0}{\pi} \right] - \operatorname{Var}_\pi \left[ \frac{\tilde{\pi}_T}{\pi} \right] = 2 \int_0^{Th} \operatorname{PFI}(\tilde{\pi}_t \| \pi) \, \mathrm{d}t.$$

The proof is then completed by noting that according to Hölder's inequality,

$$\int_{\mathcal{X}} \left\| \nabla \frac{\int_0^{Th} \tilde{\pi}_t \, \mathrm{d}t}{\pi}(x) \right\|^2 \pi(x) \, \mathrm{d}x \le \int_0^{Th} \int_{\mathcal{X}} \left\| \nabla \frac{\tilde{\pi}_t}{\pi}(x) \right\|^2 \pi(x) \, \mathrm{d}x \, \mathrm{d}t = \int_0^{Th} \operatorname{PFI}(\tilde{\pi}_t \| \pi) \, \mathrm{d}t.$$

The proof is then completed. $\square$

We then restate the main result from [19], which provides a tight characterization of the discretization error between LMC and LD.

**Proposition 3** (Theorem 2, [19]). *Assume $\nabla V$ and $\nabla^2 V$ is L-lipschtiz. Initialize $z_0$ according to Gaussian and select $h = \mathcal{O}(1/L_1)$. Then for $t \in [0, Th]$, $\operatorname{KL}(\tilde{\pi}_t \| \pi_t) = \mathcal{O}(h^2 d^2 T)$.*

It should be noticed that in the original version of [Theorem 2, [19]], the result is only stated for interger time, i.e., $t = kh$ where $k \in \{0, \cdots, T\}$. However, their proof also applies to non-integer time which leads to the above proposition.

We are now ready to prove Corollary 2.

*Proof of Corollary 2.* Combing Proposition 2 and Lemma 2, we have that

$$\bar{\tilde{\pi}}_{Th}(S)^2 \mathrm{Var}_{\pi|S}\left[\frac{\bar{\tilde{\pi}}_{Th}|S}{\pi|S}\right] \leq \frac{\pi(S)\mathrm{Var}_\pi[\frac{\tilde{\pi}_0}{\pi}]}{2Th\rho}.$$

Using Pinsker's inequality, we further obtain

$$2\bar{\tilde{\pi}}_{Th}(S)^2 D_{\mathrm{TV}}\left[\bar{\tilde{\pi}}_{Th}|S||\pi|S\right]^2 \leq \frac{\pi(S)\mathrm{Var}_\pi[\frac{\tilde{\pi}_0}{\pi}]}{2Th\rho}. \tag{4}$$

On the other hand, according to Proposition 3 and Pinsker's inequality, we obtain that for $t \in [0, Th]$,

$$D_{\mathrm{TV}}\left[\tilde{\pi}_t||\pi_t\right]^2 \leq \mathcal{O}(h^2 d^2 T),$$

and thus

$$D_{\mathrm{TV}}\left[\tilde{\pi}_t||\pi_t\right] \leq \mathcal{O}(hd\sqrt{T}).$$

Due to the convexity of $L_1$ norm, we then obtain

$$\begin{aligned}
D_{\mathrm{TV}}\left[\bar{\tilde{\pi}}_{Th}||\bar{\pi}_{Th}\right] &= \sup_A \left|\frac{1}{Th}\int_0^{Th}\tilde{\pi}_t(A) - \frac{1}{Th}\int_0^{Th}\pi_t(A)\,\mathrm{d}t\right| \\
&\leq \frac{1}{Th}\int_0^{Th}\sup_A|\tilde{\pi}_t(A) - \pi_t(A)|\,\mathrm{d}t \\
&\leq \frac{1}{Th}\int_0^{Th}D_{\mathrm{TV}}\left[\tilde{\pi}_t||\pi_t\right]\,\mathrm{d}t = \mathcal{O}(hd\sqrt{T}). \tag{5}
\end{aligned}$$

Let $A$ be a positive constant. According to Eq. (4), we have either $\bar{\tilde{\pi}}_{Th}(S) \leq A$, or $D_{\mathrm{TV}}\left[\bar{\tilde{\pi}}_{Th}|S||\pi|S\right] = \mathcal{O}\left(\frac{\sqrt{\mathrm{Var}_\pi[\frac{\tilde{\pi}_0}{\pi}]}}{A\sqrt{Th\rho}}\right)$. In the former case, combined with Eq. (5), we have that

$$\bar{\pi}_{Th}(S) \leq A + \mathcal{O}(hd\sqrt{T}).$$

In the latter case, we have that

$$D_{\mathrm{TV}}\left[\bar{\tilde{\pi}}_{Th}|S||\bar{\pi}_{Th}|S\right] \leq \mathcal{O}\left(\frac{hd\sqrt{T}}{A}\right),$$

and thus

$$D_{\mathrm{TV}}\left[\pi||\bar{\pi}_{Th}\right] \leq \mathcal{O}\left(\frac{\sqrt{\mathrm{Var}_\pi[\frac{\tilde{\pi}_0}{\pi}]}}{A\sqrt{Th\rho}}\right) + \mathcal{O}\left(\frac{hd\sqrt{T}}{A}\right).$$

The proof is completed by selecting $h = \frac{\mathrm{Var}_\pi[\frac{\tilde{\pi}_0}{\pi}]^{\frac{1}{3}}}{T^{\frac{2}{3}}\rho^{\frac{1}{3}}d^{\frac{2}{3}}}$, and $A = \frac{d^{\frac{1}{6}}\mathrm{Var}_\pi[\frac{\tilde{\pi}_0}{\pi}]^{\frac{1}{6}}}{\rho^{\frac{1}{6}}T^{\frac{1}{12}}}$. $\qquad\square$

### A.3 Difficulty of extension to Rényi divergence

Given two distribution $\pi$ and $\mu$, Rényi divergence of $q$ between them, i.e., $\mathcal{R}_q(\mu||\pi)$ is defined as follows:

$$\mathcal{R}_q(\mu||\pi) \triangleq \frac{1}{q-1}\ln \mathbb{E}_{x\sim\pi}\left(\frac{\mu}{\pi}(x)\right)^q.$$

For simplicity, here we consider Langevin Dynamics defined as Eq. (1) with the distribution of time $t$ denoted as $\tilde{\pi}_t$. The measure of convergence is then defined as $\mathcal{R}_q(\pi_t||\pi)$, the derivative of which, according to (Lemma 6, [23]) is given as

$$\frac{d}{dt}\mathcal{R}_q(\pi_t||\pi) = -q\frac{G_q\left(\pi_t||\pi\right)}{F_q\left(\pi_t||\pi\right)},$$

where

$$G_q\left(\pi_t||\pi\right) = \mathbb{E}_\pi\left[\left(\frac{\pi_t}{\pi}\right)^q\left\|\nabla\log\frac{\pi_t}{\pi}\right\|^2\right] = \mathbb{E}_\pi\left[\left(\frac{\pi_t}{\pi}\right)^{q-2}\left\|\nabla\frac{\pi_t}{\pi}\right\|^2\right] = \frac{4}{q^2}\mathbb{E}_\pi\left[\left\|\nabla\left(\frac{\pi_t}{\pi}\right)^{\frac{q}{2}}\right\|^2\right],$$

$$F_q\left(\pi_t||\pi\right) = \mathbb{E}_\pi\left[\left(\frac{\pi_t}{\pi}\right)^q\right].$$

If we would like to follow the routine of Lemma 1, we need to firstly lower bound $\frac{G_q(\pi_t||\pi)}{F_q(\pi_t||\pi)}$ by $\frac{G_q((\pi_t|S)||(\pi|S))}{F_q((\pi_t|S)||(\pi|S))}$ and secondly transfer $\frac{G_q((\pi_t|S)||(\pi|S))}{F_q((\pi_t|S)||(\pi|S))}$ back to $\mathcal{R}_q((\pi_t|S)||(\pi|S))$. The second step can be accomplished following the same routine as (Lemma 9,[24]) using local PI. However, we are still unaware of how to implement the first step. This is because although we have

$$G_q\left(\pi_t||\pi\right) = \frac{4}{q^2}\mathbb{E}_\pi\left[\left\|\nabla\left(\frac{\pi_t}{\pi}\right)^{\frac{q}{2}}\right\|^2\right] \geq \frac{4}{q^2}\mathbb{E}_{\pi|S}\left[\left\|\nabla\left(\frac{\pi_t|S}{\pi|S}\right)^{\frac{q}{2}}\right\|^2\right] \times \frac{\pi_t(S)^q}{\pi(S)^{q-1}},$$

which is similar to the analysis of local LSI, we also have

$$F_q\left(\pi_t||\pi\right) = \mathbb{E}_\pi\left[\left(\frac{\pi_t}{\pi}\right)^q\right] \geq \mathbb{E}_\pi\left[\left(\frac{\pi_t}{\pi}\right)^q\right] \times \frac{\pi_t(S)^q}{\pi(S)^{q-1}}.$$

In other words, when restricting to $S$, both the denominator and the numerator of $\frac{G_q(\pi_t||\pi)}{F_q(\pi_t||\pi)}$ will get smaller, which makes $\frac{G_q(\pi_t||\pi)}{F_q(\pi_t||\pi)}$ no longer lower bounded by $\frac{G_q((\pi_t|S)||(\pi|S))}{F_q((\pi_t|S)||(\pi|S))}$. We leave how to resolve this challenge as an interesting future work.

# B  Proof of Applications

## B.1  Proof of gaussian-mixture

To start with, we recall Bakry-Émery criterion and Holley-Stroock perturbation principle.

**Lemma 8** (Bakry-Émery criterion). *Let $\Omega \subset \mathbb{R}^n$ be convex and let $H : \Omega \to \mathbb{R}$ be a Hamiltonian with Gibbs measure $\mu(x) \propto e^{-H(x)}1_\Omega(x)$ and assume that $\nabla^2 H(x) \geq \kappa > 0$ for all $x \in supp(\mu)$. Then $\mu$ satisfies $\mathrm{LSI}(\kappa)$.*

**Lemma 9** (Holley-Stroock perturbation principle). *If $p \in \Delta(\Omega)$ satisfies $\mathrm{LSI}(\pi_t)$, and $\psi : \Omega \to \mathbb{R}$ satisfies $m \leq \psi \leq M$, where $m, M > 0$. Then, $q \in \Delta(\Omega) \propto \psi p$ satisfies $\mathrm{LSI}(\frac{m}{M}\pi_t)$.*

We are now ready to prove Lemma 3.

*Proof of Lemma 3.* Since $S_i$ is convex, by applying Lemma 8, we obtain $p_i|_{S_i}$ is $\mathrm{LSI}(\sigma^2)$. Let $c = \min_i w_i$.

Meanwhile, we have over $S_j$

$$cp_j \leq w_j p_j \leq p = \sum_{i\neq j} w_i p_i + w_j p_j \leq \sum_{i\neq j} w_i p_j + w_j p_j = p_j.$$

Therefore, by Holley-Stroock perturbation principle, we have that $p|_{S_j}$ satisfies $\mathrm{LSI}(\frac{1}{c}\sigma^{-2})$. $\square$

*Proof of Lemma 4.* Through direct calculation, we obtain

$$\nabla^2 V = \Sigma^{-1} - \frac{1}{2}\Sigma^{-1}\frac{\sum_{i,j} w_i w_j p_i(x) p_j(x)(\mu_i - \mu_j)(\mu_i - \mu_j)^\top}{(\sum_{i=1}^n w_i p_i(x))^2}\Sigma^{-1}.$$

Then, for any $\boldsymbol{a} \in \mathbb{R}^d$ with $\|\boldsymbol{a}\| = 1$, we have

$$\boldsymbol{a}^\top \nabla^2 V \boldsymbol{a} = \boldsymbol{a}^\top \Sigma^{-1}\boldsymbol{a} - \frac{1}{2}\frac{\sum_{i,j} w_i w_j p_i(x) p_j(x)|(\mu_i - \mu_j)^\top \Sigma^{-1}\boldsymbol{a}|^2}{(\sum_{i=1}^n w_i p_i(x))^2}$$

$$\leq \boldsymbol{a}^\top \Sigma^{-1}\boldsymbol{a} \leq \frac{1}{\sigma^{-2}},$$

and

$$
\begin{aligned}
\boldsymbol{a}^\top \nabla^2 V \boldsymbol{a} \leq &\, \boldsymbol{a}^\top \Sigma^{-1} \boldsymbol{a} - \frac{1}{2} \frac{\sum_{i,j} w_i w_j p_i(x) p_j(x) |(\mu_i - \mu_j)^\top \Sigma^{-1} \boldsymbol{a}|^2}{(\sum_{i=1}^n w_i p_i(x))^2} \\
\geq &\, -\frac{1}{2} \frac{\sum_{i,j} w_i w_j p_i(x) p_j(x) |(\mu_i - \mu_j)^\top \Sigma^{-1} \boldsymbol{a}|^2}{(\sum_{i=1}^n w_i p_i(x))^2} \geq -\frac{\max_{i,j} \|\mu_i - \mu_j\|^2}{\sigma^{-4}}.
\end{aligned}
$$

The proof is completed. □

## B.2  Proof of sampling from the power posterior distribution of symmetric Gaussian mixtures

To begin with, define the expected potential function $\bar{V}$ as

$$
\bar{V}(\theta) := \beta \mathbb{E}_X \log \left( \frac{1}{2} \varphi \left( \theta - X \right) + \frac{1}{2} \varphi \left( \theta + X \right) \right) + \log \lambda(\theta).
$$

We further define the corresponding distribution as $\bar{\pi} \propto e^{-\bar{V}}$.

The following lemmas will be needed in the proof.

**Lemma 10** (Theorem 2, [20])**.** *If $A, M \geq \|\|\theta_0\|\| + 1$, we have $\bar{\pi}|_{R_1}$ satisfies $\mathrm{PI}(\Theta(1/A^4 M^2))$.*

**Lemma 11** (Lemma 4, [20])**.** *If $A, M \geq \|\theta_0\| + 1$, we have with probability at least $1 - \delta$,*

$$
\sup_{\theta \in [0,A] \times \mathcal{B}(0,M)} |V(\theta) - \bar{V}(\theta)| \leq \mathcal{O}((1 + A + M) \sqrt{\frac{d}{n} \log \frac{n(A + M + d)}{\delta}}).
$$

**Lemma 12.** *Suppose $V$ obeys strong dissipativity, i.e., $\nabla V$ is $L$-lipschitz and $\langle \nabla V(x), x \rangle \geq m\|x\|^2 - b$. Then, running LMC with $h < \frac{1}{16L}$ and $h \leq \frac{8m}{4b+32d}$, we have*

$$
\mathbb{E}_{x_k \sim \pi_k} e^{\frac{1}{4m} \|x_k\|^2} \leq 32 \cdot \exp \left( \frac{1}{4m} \left( 8b + 64d \right) \right).
$$

*Proof.* Define $f(x) = e^{\frac{1}{4m} \|x\|^2}$. Assume wlog that $\nabla V(0) = 0$.

Then,

$$
\begin{aligned}
f(x_{k+1}) &= \exp \left( \frac{1}{4m} \|x_k\|^2 + \frac{1}{2m} \left\langle -h \nabla V(x_k) + \sqrt{2h} \xi_k, x_k \right\rangle + \frac{1}{4m} \left\| -h \nabla V(x_k) + \sqrt{2h} \xi_k \right\|^2 \right) \\
&\leq \exp \left( \frac{1}{4m} \|x_k\|^2 - \frac{h}{2m} \left( \|x_k\|^2 - b \right) + \frac{\sqrt{2h}}{2m} \langle \xi_k, x_k \rangle + \frac{1}{2m} \left\| h^2 \nabla V(x_k) \right\|^2 + \frac{h}{m} \|\xi_k\|^2 \right) \\
&\leq \exp \left( \frac{1}{4m} \|x_k\|^2 - \frac{h}{4m} \left( \|x_k\|^2 - 2b \right) + \frac{\sqrt{2h}}{2m} \langle \xi_k, x_k \rangle + \frac{h}{m} \|\xi_k\|^2 \right).
\end{aligned}
$$

Let $\mathbb{E}_k$ denote expectation wrt $\xi_k$. Then

$$
\begin{aligned}
\mathbb{E}_k[f(x_{k+1})] &\leq \exp \left( \frac{1}{4m} \|x_k\|^2 - \frac{h}{4m} \left( \|x_k\|^2 - 2b \right) \right) \cdot \mathbb{E}_k \left[ \frac{\sqrt{2h}}{2m} \langle \xi_k, x_k \rangle + \frac{h}{m} \|\xi_k\|^2 \right] \\
&\leq \exp \left( \frac{1}{4m} \|x_k\|^2 - \frac{h}{4m} \left( \|x_k\|^2 - 2b \right) \right) \cdot \mathbb{E}_k \left[ \frac{\sqrt{2h}}{m} \langle \xi_k, x_k \rangle \right]^{1/2} \cdot \mathbb{E}_k \left[ \frac{2h}{m} \|\xi_k\|^2 \right]^{1/2}
\end{aligned}
$$

Using the fact that $\chi^2$ variable is sub-exponential,

$$
\mathbb{E}_k \left[ \exp \left( \frac{2h}{m} \|\xi_k\|^2 \right) \right] \leq \exp \left( \frac{4hd}{m} \right)
$$

On the other hand, notice that $\langle \xi_k, x_k \rangle \sim \mathcal{N}(0, \|x_k\|^2)$ is a 1-dimensional gaussian random variable. It can be shown that

$$\mathbb{E}_k \left[ \exp \left( \frac{\sqrt{2h}}{m} \langle \xi_k, x_k \rangle \right) \right] \leq \exp \left( \frac{2h}{m^2} \|x_k\|^2 \right). \tag{6}$$

Combining the above,

$$\mathbb{E}_k[f(x_{k+1})] \leq \exp \left( \frac{1}{4m} \|x_k\|^2 - \frac{h}{4m} \left( \|x_k\|^2 - 2b \right) + \frac{2hd}{m} + \frac{h}{m^2} \|x_k\|^2 \right)$$

$$\leq \exp \left( \frac{1}{4m} \|x_k\|^2 - \frac{h}{8m} \left( \|x_k\|^2 - 4b - 32d \right) \right)$$

Let $\mathbb{E}$ denote expectation wrt all randomness. Then

$\mathbb{E}[f(x_{k+1})]$

$\leq \mathbb{E} \left[ \exp \left( \frac{1}{4m} \|x_k\|^2 - \frac{h}{8m} \left( \|x_k\|^2 - 4b - 32d \right) \right) \right]$

$= \mathbb{E} \left[ \exp \left( \frac{1}{4m} \|x_k\|^2 - \frac{h}{8m} \left( \|x_k\|^2 - 4b - 32d \right) \right) 1 \left\{ \|x_k\|^2 \geq 8b + 64d \right\} \right]$

$\quad + \mathbb{E} \left[ \exp \left( \frac{1}{4m} \|x_k\|^2 - \frac{h}{8m} \left( \|x_k\|^2 - 4b - 32d \right) \right) 1 \left\{ \|x_k\|^2 \leq 8b + 64d \right\} \right]$

$\leq \mathbb{E} \left[ \exp \left( \frac{1}{4m} \|x_k\|^2 - \frac{h}{16m} \left( 4b + 32d \right) \right) 1 \left\{ \|x_k\|^2 \geq 8b + 64d \right\} \right]$

$\quad + \mathbb{E} \left[ \exp \left( \frac{1}{4m} \|x_k\|^2 + \frac{h}{8m} \left( 4b + 32d \right) \right) 1 \left\{ \|x_k\|^2 \leq 8b + 64d \right\} \right]$

$\leq \mathbb{E} \left[ \exp \left( \frac{1}{4m} \|x_k\|^2 \right) 1 \left\{ \|x_k\|^2 \geq 8b + 64d \right\} \cdot \left( 1 - \frac{h}{32m} \left( 4b + 32d \right) \right) \right]$

$\quad + \mathbb{E} \left[ \exp \left( \frac{1}{4m} \|x_k\|^2 \right) 1 \left\{ \|x_k\|^2 \leq 8b + 64d \right\} \cdot \left( 1 + \frac{h}{4m} \left( 4b + 32d \right) \right) \right]$

$= \mathbb{E} \left[ \exp \left( \frac{1}{4m} \|x_k\|^2 \right) \right] - \frac{h}{32m} \left( 4b + 32d \right) \mathbb{E} \left[ \exp \left( \frac{1}{4m} \|x_k\|^2 \right) 1 \left\{ \|x_k\|^2 \geq 8b + 64d \right\} \right]$

$\quad + \frac{h}{4m} \left( 4b + 32d \right) \mathbb{E} \left[ \exp \left( \frac{1}{4m} \|x_k\|^2 \right) 1 \left\{ \|x_k\|^2 \leq 8b + 64d \right\} \right]$

$\leq \mathbb{E} \left[ \exp \left( \frac{1}{4m} \|x_k\|^2 \right) \right] - \frac{h}{32m} \left( 4b + 32d \right) \mathbb{E} \left[ \exp \left( \frac{1}{4m} \|x_k\|^2 \right) \right]$

$\quad + \frac{h}{m} \left( 4b + 32d \right) \cdot \exp \left( \frac{1}{4m} \left( 8b + 64d \right) \right)$

$= \mathbb{E}[f(x_k)] - \frac{h}{32m} \left( 4b + 32d \right) \mathbb{E}[f(x_k)] + \frac{h}{m} \left( 4b + 32d \right) \cdot \exp \left( \frac{1}{4m} \left( 8b + 64d \right) \right).$

Suppose that $x_k$ is drawn from the invariant distribution under LMC. Then we know that $\mathbb{E} f(x_k) = \mathbb{E} f(x_{k+1})$. In this case,

$$0 \leq -\frac{h}{32m} \left( 4b + 32d \right) \mathbb{E}[f(x_k)] + \frac{h}{m} \left( 4b + 32d \right) \cdot \exp \left( \frac{1}{4m} \left( 8b + 64d \right) \right).$$

Moving things around gives

$$\mathbb{E} f(x_k) \leq 32 \cdot \exp \left( \frac{1}{4m} \left( 8b + 64d \right) \right).$$

$\square$

We are now ready to prove the lemmas in the main text.

*Proof of Lemma 5.* Based on Lemma 11, if $n \geq \tilde{\Theta}((A + M)^2 d \log(1/\delta))$, we have with probability at least $1 - \delta$,

$$\sup_{\theta \in R_1} |V(\theta) - \bar{V}(\theta)| \leq \mathcal{O}(1).$$

As a result, we have $\frac{\pi|R_1}{\pi|R_1}(\theta) = \Theta(1)$, and by Lemma 9 and Lemma 10, the proof is completed. □

*Proof of Lemma 6.* To begin with, set $\xi \sim \mathcal{N}(0, \mathbb{I}_d)$, we have

$$\langle \nabla \bar{V}(\theta), \theta \rangle = \beta \|\theta\|^2 + \beta \mathbb{E} \left( \frac{-\varphi(\|\theta_0\|e_1 + \xi - \theta) + \varphi(\|\theta_0\|e_1 + \xi + \theta)}{\varphi(\|\theta_0\|e_1 + \xi - \theta) + \varphi(\|\theta_0\|e_1 + \xi + \theta)} \theta^\top (\|\theta_0\|e_1 + \xi) \right)$$

$$\geq \frac{\beta}{2} \|\theta\|^2 - \beta \left( \|\theta_0\|^2 + 1 \right).$$

We obtain

$$\langle \nabla V(\theta), \theta \rangle \geq \frac{\beta}{2} \|\theta\|^2 - 2\beta \left( \|\theta_0\|^2 + 1 \right)$$

following a standard empirical process argument due to $n \geq \tilde{\Theta}((A + M)^2 d \log(1/\delta))$ (see Lemma 6, [20] as an example).

Meanwhile, denote $\tau_i = 1$ if $X_i$ is sampled from $\mathcal{N}(\theta_0, \mathbb{I}_d)$ and $\tau_i = -1$ else-wise. We have

$$\nabla^2 V(\theta) = \beta \mathbb{I}_d - \frac{\|\theta_0\|^2 \beta}{n} \sum_{i=1}^n \left( \frac{4\varphi(X_i - \theta)\varphi(X_i + \theta)}{(\varphi(X_i - \theta) + \varphi(X + \theta))^2} \right) e_1 e_1^\top$$

$$- \frac{\beta}{n} \sum_{i=1}^n \left( \frac{4\varphi(X_i - \theta)\varphi(X_i + \theta)}{(\varphi(X_i - \theta) + \varphi(X_i + \theta))^2} (X_i - \tau_i \|\theta_0\|e_1)(X_i - \tau_i \|\theta_0\|e_1)^\top \right),$$

and thus $(\beta - \|\theta_0\|^2 \beta - \frac{\beta}{n} \sum_{i=1}^n \|X_i - \tau_i \|\theta_0\|e_1\|^2) \mathbb{I}_d \leq \nabla^2 V(\theta) \leq \beta \mathbb{I}_d$. As $X_i - \tau_i \|\theta_0\|e_1 \sim \mathcal{N}(0, \mathbb{I}_d)$, by concentration inequality of chi-square variable and since $n \geq \tilde{\Theta}((A + M)^2 d \log(1/\delta))$, we have with probability at least $1 - \delta$,

$$\|\nabla^2 V(\theta)\| \leq 2\beta(1 + \|\theta_0\|^2).$$

The claim for $\|\nabla^3 V\|$ can be calculated following the similar routine. The proof is completed. □

*Proof of Lemma 7.* The claim directly follows by applying Markov's inequality to Lemma 12. □

## C  Proof of Theorem 1

*Proof.* We consider the variational form of spectral gap. We first define the Dirichlet form and the variance

$$\mathcal{E}_P(\phi, \phi) = \langle \phi, (I - P)\phi \rangle_\pi = \sum_{x,y} \pi(x)\phi(x)(\mathbb{I}(x, y) - P(x, y))\phi(y),$$

$$\text{Var}_\pi[\phi] = \sum_x \pi(x)(\phi(x) - \mathbb{E}_\pi[\phi])^2$$

Then by the fact that $P$ is lazy, and hence $P = \frac{1}{2}(I + \hat{P})$ for some reversible transition $\hat{P}$, we have

$$\text{Var}_\pi[P\phi] \leq \text{Var}_\pi[\phi] - \mathcal{E}_P(\phi, \phi),$$

By nonnegativity $\mathcal{E}_P(\phi, \phi) \geq 0$,, we have

$$\sum_{t \leq T} \mathcal{E}_P \left( \frac{\mu_t}{\pi}, \frac{\mu_t}{\pi} \right) \leq \text{Var}_\pi \left[ \frac{\mu_0}{\pi} \right]. \tag{7}$$

By the variational form of spectral gap,

$$\alpha \leq \inf_{\phi \text{ non-constant}} \frac{\mathcal{E}_P(\phi, \phi)}{\text{Var}_\pi[\phi]}.$$

Therefore, applying the Markov chain generated by $P_i$ we have that for any $i \leq m$,

$$\sum_t \text{Var}_{\pi_i}\left[\frac{\mu_t | \mathcal{X}_i}{\pi | \mathcal{X}_i}\right] \leq \frac{1}{\alpha} \sum_t \mathcal{E}_{P_i}\left(\frac{\mu_t | \mathcal{X}_i}{\pi | \mathcal{X}_i}, \frac{\mu_t | \mathcal{X}_i}{\pi | \mathcal{X}_i}\right) \tag{8}$$

We then note that

$$\mathcal{E}_P\left(\frac{\mu_t}{\pi}, \frac{\mu_t}{\pi}\right) = \frac{1}{2} \sum_{x,y} \pi(x) P(x, y) (\frac{\mu_t}{\pi}(x) - \frac{\mu_t}{\pi}(y))^2$$

$$\geq \frac{1}{2} \sum_{x,y \in \mathcal{X}_i} \pi(x) P(x, y) (\frac{\mu_t}{\pi}(x) - \frac{\mu_t}{\pi}(y))^2$$

$$= \frac{1}{2} \sum_{x,y \in \mathcal{X}_i} \pi(x) P_i(x, y) (\frac{\mu_t}{\pi}(x) - \frac{\mu_t}{\pi}(y))^2$$

$$+ \frac{1}{2} \sum_{x,y \in \mathcal{X}_i} \pi(x) (P(x, y) - P_i(x, y)) (\frac{\mu_t}{\pi}(x) - \frac{\mu_t}{\pi}(y))^2$$

We note by (PI) that for any $x, y \in \mathcal{X}_i, x \neq y$, $P(x, y) = P_i(x, y)$ Therefore, the second term is zero. Hence,

$$\mathcal{E}_P\left(\frac{\mu_t}{\pi}, \frac{\mu_t}{\pi}\right) \geq \frac{\pi(\mathcal{X}_i)^2}{2\mu_t(\mathcal{X}_i)^2} \mathcal{E}_{P_i}\left(\frac{\mu_t | \mathcal{X}_i}{\pi | \mathcal{X}_i}, \frac{\mu_t | \mathcal{X}_i}{\pi | \mathcal{X}_i}\right)$$

Combining with (7) and (8) we get,

$$\frac{1}{T} \sum_t \text{Var}_{\pi_i}\left[\frac{\mu_t | \mathcal{X}_i}{\pi | \mathcal{X}_i}\right] / \mu_t(\mathcal{X}_i)^2 \leq \frac{2}{\pi(\mathcal{X}_i)^2 \alpha T} \text{Var}_\pi\left[\frac{\mu_0}{\pi}\right] \tag{9}$$

The claim then follows by discussing if for all $t$, $\mu_t(\mathcal{X}_i) \leq \pi(\mathcal{X}_i) T^{-1/4}$, then

$$\frac{1}{T} \sum_t \text{Var}_{\pi_i}\left[\frac{\mu_t | \mathcal{X}_i}{\pi | \mathcal{X}_i}\right] \leq \frac{2}{\alpha T^{1/2}} \text{Var}_\pi\left[\frac{\mu_0}{\pi}.\right] \tag{10}$$

$\square$

## D   Proof for Theorem 2

*Proof.* Given the subgraph $V$, we have that conditioned transition kernel is

$$p'(x, y) = p(x, y) + \mathbb{1}\{x = y\} P(x, (V')^c).$$

We analyze the conductance on graph $V'$ induced by $p'$,

$$\Phi = \min_{V_1 \subset V'} \frac{\sum_{x \in V_1} \pi(x) p'(x, V_1^c)}{\min\{\pi(V_1), 1 - \pi(V_1)\}}$$

For any partition $V_1, V_2$ of $V'$, without loss of generality assume the local min of $f$ is in $V_2$, $v^* \in V_2$. For simplicity, for any $x \in V_1$, we denote $\tau(x)$ be the last element in its shortest path to $v^*$. We note that

$$d(x, \tau(x)) \leq D, \pi(x) \leq \pi(\tau(x)).$$

Denote $\tau(V_1)$ be the set of such elements in $V_1$. Then we have

$$\sum_{x \in V_1} \pi(x) p'(x, V_1^c) \geq \frac{\sum_{x \in \tau(V_1)} \pi(x)}{4d}. \tag{11}$$

Further denote $\tau(x, r)$ for some $r \in \{0, 1, ..., d\}$ as the set of elements at distance $r$ from $\tau(x)$ along the descending shortest paths. We provide an illustrative figure in 2. Then we have that

$$\pi(V_1) = \sum_{\tau(x) \in \tau(V_1)} \sum_{r=1}^{D} \tau(x, r). \tag{12}$$

We note by reversibility that

$$\pi(\tau(x, r)) p'(\tau(x, r), \tau(x, r+1)) = \tau(x, r+1) p'(\tau(x, r+1), \tau(x, r)). \tag{13}$$

Further by the fact that the function along the path is descending, we get that

$$p'(\tau(x, r), \tau(x, r+1)) \geq p'(\tau(x, r+1), \tau(x, r)). \tag{14}$$

Therefore,

$$\pi(\tau(x, r)) \geq \pi(\tau(x, r+1)). \tag{15}$$

Hence we have

$$\pi(V_1) = \sum_{\tau(x) \in \tau(V_1)} \sum_{r=1}^{D} \tau(x, r) \leq D\pi(\tau(V_1)) \tag{16}$$

Therefore, when $\pi(V_1) \leq 1/2$ we have

$$\frac{\sum_{x \in V_1} \pi(x) p'(x, V_1^c)}{\min\{\pi(V_1), 1 - \pi(V_1)\}} \geq \frac{1}{4dD}.$$

when $\pi(V_1) \geq 1/2$ we have

$$\frac{\sum_{x \in V_1} \pi(x) p'(x, V_1^c)}{\min\{\pi(V_1), 1 - \pi(V_1)\}} \geq 2 \sum_{x \in V_1} \pi(x) p'(x, V_1^c)$$

$$\geq \frac{\sum_{x \in \tau(V_1)} \pi(x)}{2d} \geq \frac{\pi(\tau(V_1))}{2dD} \geq \frac{1}{4dD}.$$

The theorem then follows by Cheeger's inequality.

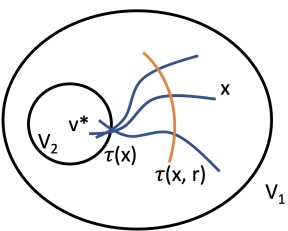

Figure 2: An illustration for the definitions of $\tau(x)$ and $\tau(x, r)$.

$\square$

# E    Additional Experiments

We still use $\pi_3$ in 6.1 as our target distribution. We initially set $n = 1, 10, 2000$ particles to run the LMC sampling algorithm respectively. We collect the locations of these particles after $T = 1000$ iterations as valid samples. The target distribution is shown in Figure 3a; empirical distributions using $n = 1, 10, 2000$ particles are shown in Figure 3b, 3c, and 3d.

We highlight two observations from the experiments. *First* when we use only one particle, it is always trapped in one Gaussian component and we never get samples from other modes. It has very bad global convergence, but still gets good conditional convergence: The convergence result conditioned

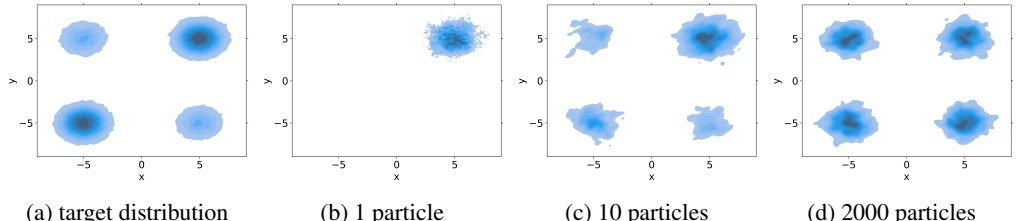

| (a) target distribution | (b) 1 particle | (c) 10 particles | (d) 2000 particles |

on the 1st quadrant is good, and the sample after $T$ iterations has no probability distributed on the other 3 quadrants. *Second*, when the number of particles grows, the sample after $T$ iterations has non-negligible probabilities distributed on all of the 4 quadrants, which means we could capture all of the Gaussian components. The results indicate that although restarting LMC may not directly lead to global convergence, it may help us capture the features in multi-modal distributions, which further implies we may empirically eliminate the "small probability mass" condition in Corollary 1 and mitigate the gap between sampling distribution and target distribution.

A broader implication could be on ensemble and stochastic averaging in neural network training, where ensemble follows the restart procedure where as stochastic averaging is closer to sampling from a single trajectory. Our theory and experiment suggest that the two can have very different distributions in the end.

