# OpenReview forum: "Fast Conditional Mixing of MCMC Algorithms for Non-log-concave Distributions"
_NeurIPS.cc/2023/Conference — NeurIPS 2023 poster_

### Official Review · Reviewer_Lt2V · 2023-06-09

**Soundness:** 2 fair
**Presentation:** 2 fair
**Contribution:** 3 good
**Rating:** 7
**Confidence:** 4

**Summary:**

This paper introduces a new concept of conditional convergence of an algorithm (i.e. convergence of the distribution restricted to e.g. a local mode), and presents extension of recent results using e.g. Poincare Inequality to bound this new measure.

**Strengths:**

I think this paper tackles an important problem: sampling from multi-modal distributions is likely to get stuck in a local optima and so understanding how well it samples from that mode is sensible (particularly when thinking about how a sampling based approach compares to an optimisation one).

The idea and definition of conditional convergence is new and interesting -- and one could see a new body of research around this type of topic.

The theory is based on a single, simple results (Lemma 1) which can then be used to generalise existing results on convergence to conditional convergence.

**Weaknesses:**

The main weakness is that the current result, e.g. Corollary 1, are weaker than one would hope. In particular

Corollary 1 seems to have a time-dependent step-size (so to understand pi_T we need a smaller step size as T increases). This is different from how algorithms are implemented.

The conclusion after corollary 1 does not make sense to me. “Either the probability mass … over S converges to 0, or…” : But whilst LMC can converge slowly, it still converges to pi (if we ignore the discretisation error). Thus surely for any set S that is not a null-set of pi we have pi_T(S) will be non-zero in the limit as T goes to infinity?

More generally, this corollary is quite the result you want. You want something that says “if I have a partition of the the state-space in sets S_1,S_2,\ldots, and observe that the proportion of time LMC stays in S_i is greater than … then the LMC output restricted to S_i will be close to pi restricted to S_i” or similar.

So I struggle to really understand the practical importance of the result.

Separately, the presentation could be improved in places:

E.g. missing “the”s for *the* local Poincare inequality, and *the* Poincare Figure information (and e.g. “fisher”->”Fisher”, gaussian to Gaussian etc.); also “Gibbs sampling on discrete state *spaces*” etc.

When describing Langevin Monte Carlo, there are two different approaches depending on whether you use a Metropolis correction. The paper could be clearer about what its results relate to. Form the definition in Algorithm 1 it looks like you are considering the unadjusted Langevin algorithm (ULA).

If you are using ULA, then there is a discretisation error — i.e. the stationary distribution for ULA is different to that of the Langevin diffusion/the distribution you want to sample from. I think the paper could be up-front about this, and discuss how this impacts the results in the paper. I also wonder if informal statements such as “The convergence of LMC” (top p.4) are really referring to the convergence of the Langevin diffusion and not ULA. The paper does comment on “PI .. implies the convergence of LMC when the step size h is small”, but this could be more precise. I guess part of the confusion is that you are commenting on converging, but it is not completely clear what you are converging to. In general the ULA will not converge to pi for a fixed (albeit small) step-size as the discretisation error will change the stationary distribution.

The argument under Fisher information seems to ignore the discretisation error — i.e. be just for the Langevin diffusion, but is says for LMC. (Though this is corrected in Proposition 1 which is for LMC). This part could have been improved with some references to the results you are summarising.

[It may be that a cleaner presentation would be to have had a subsection commenting on the approximation error between ULA and the Langevin diffusion, and then presenting the results just for the Langevin diffusion?]



**Questions:**

Can you more convincingly explain the practical importance of Corollary 1?

As part of this, it feels like the rate of convergence matters (as we have convergence of LMC in general, just that it can be very slow). The slowness of LMC is, I believe, linked to e.g. constants for convergence depending on features of the target (more separated modes implies slower convergence) -- so it feels like you want stronger results than Corollary 1 where you get a handle on the constants. I.e. are the results in Corollary 1 uniform over targets?

---

> ### Author Rebuttal · Authors · 2023-08-09
>
> We thank the reviewer for acknowledging the importance of our problem. After reading the review, we find that there could be several misunderstandings. We begin with clarifying these misunderstandings and then address other concerns.
>
> **On "convergence of LMC".** In this paper, when we talk about "convergence of LMC", we mean that "by running LMC with step size $h$ and total iteration $T$, we can obtain a distribution $\mu$ with distance (global or conditional) smaller than $\varepsilon$ from the target distribution $\pi$, where $h$ and $T$ can depend on $\varepsilon$". This definition of convergence has been employed by many existing works including [1,2]. The reviewer seems to misunderstand the "convergence" as "the step size $h$ does not depend on $\varepsilon$". We will include the above explanation of "convergence of LMC" in the revised paper.
>
>
> **On  Practical Importance of Corollary 1.** To address this point, we briefly recall the motivation and goal of our study:
>
> (1)The global convergence of LMC in general non-log-concave cases has been proven to be **exponentially slow in the worst case**. For a simple example, consider the following mixture of two standard Gaussians:
> > $p(x;c) \propto \frac{1}{2} e^{- (x+c)^2} + \frac{1}{2} e^{- (x-c)^2}$.
>
> The global convergence is provably exponentially slow in the separation distance $c$.
>
> (2) However, Corollary 1 shows the remarkable fact that **local mixing can be fast even when global mixing is provably slow.** For instance, for $p(x;c)$ above, $\pi_t \vert_{(-\infty, 0]}$  is $\epsilon$ close to $p(x;c) \vert_{(-\infty, 0]}$ in $O(1/\varepsilon^4)$ steps, **independent of $c$** (ditto for $[0,\infty)$). As the reviewer notes, **one would expect that** "more separated modes imply slower convergence", so it is highly remarkable that local convergence rate **in fact does not depend on the separation distance $c$!**
>
> More generally, Corollary 1 guarantees polynomial-time sampling as long as **each mixture component has a good LSI constant**, which is a much more lax requirement than for the entire distribution to have good LSI. For instance, each component of $p(x;c)$ has LSI constant of $\Theta(1)$, but $p(x;c)$ has LSI constant of $\Theta(e^{-c^2})$.
>
> (3) The catch to Corollary 1 is that local convergence says nothing about mixture weights, so you can have two very unbalanced modes. However, in many practical settings, even guaranteeing accurate samples within a mode is valuable (e.g. one might care more about generating accurate pictures of dogs/cats, and care less about the ratio of dog pictures to cat pictures.)
>
>
> **Other concerns.**
>
> **Q1**: Corollary 1 uses time-dependent step-size.
>
> **A1**: In all our results, T stands for the total iterations that the algorithm runs. The T-dependent stepsize can be easily transformed into an epsilon-dependent step-size. For instance, in Corollary 1, for epsilon target accuracy we need $T = O(\alpha^2 d^2 /\epsilon^4)$, so the stepsize $\delta = \alpha d/\epsilon^2$. Perhaps you will find this formulation more familiar, and most MCMC papers do have stepsize depending on epsilon (e.g. in [1,2]).
>
> **Q2**： "probability mass ... converges to 0" seems redundant.
>
> **A2**: The reviewer also acknowledges that "LMC can converge slowly". In fact, the global convergence rate for a simple mixture of Gaussians can often be as slow as $e^{-d}$. It is therefore highly non-trivial that Corollary 1 has polynomial dependence on all problem parameters. As T goes to infinity, a "set S that is not a null-set of pi" will indeed eventually have significant mass, but this can take **an exponentially long time (see the Gaussian mixture example in "On  Practical Importance of Corollary 1." above)**, so one cannot simply assume that every set S has sufficiently positive probability in practical settings when the total iteration number T is not exponentially in problem dimension d.
>
> **Q3**：" this corollary is quite the result you want"
>
> **A3** The reviewer's suggestion is in fact quite similar to our Corollary 1. Consider the following re-statement of Corollary 1, using similar phrasing as the reviewer's suggestion:
> > “if I have a partition of the the state-space in sets S_1,S_2,\ldots, and observe that **the probability $Z_t$ stays in S_i** is greater than … then **$\bar{\pi_t}$ restricted to S_i** will be close to $\pi$ restricted to S_i”
>
> The difference from the reviewer's suggestion is bolded. Qualitatively, Corollary 1 is almost identical to the reviewer's suggestion, except we consider probability at time $t$ ($\mu_t$) instead of path-average. We are thus unsure about what the reviewer means here. We will greatly appreciate it if the reviewer can elaborate on/clarify this point.
>
>
> **Q4**: Presentation.
>
> **A4**: We thank the reviewer for this suggestion and will correct the typos and update the manuscript. We will also further highlight (currently in line 113-118) that we consider the unadjusted Langevin algorithm and the analyses done in our work are for discretized algorithms.
>
>
> **Q5**: Is Corollary 1 uniform over all targets?
>
> **A5**: We are uncertain about the reviewer's use of the term "uniform," but we will attempt to clarify it in this context.
>
> 1. It is indeed possible to derive outcomes from Corollary 1 based on certain "features of the target." This is due to the rate in Corollary 1 being dependent on the local LSI constant $\alpha$, which in turn relies on the "features." In Section 4.3, we demonstrate how to calculate the local LSI using the "features."
>
> 2. Although the global convergence rate is also influenced by the "features," it can be exponentially slow and, therefore, impractical. This is exemplified by the mixture of two standard Gaussian distributions, as discussed earlier.
>
>
> **References:**
>
> [1].  Analysis of Langevin Monte Carlo from Poincar´e to Log-Sobolev
>
> [2]. Towards a Theory of Non-Log-Concave Sampling: First-Order Stationarity Guarantees for Langevin Monte Carlo

---

> > ### Comment · Reviewer_Lt2V · 2023-08-16
> >
> > My concern about Corollary 1 is that the narrative after the corollary was that either the chain will have small probability to S or the conditional distribution (given S) is well approximated. But this is not true. As an example consider the your mixture example for large c and a chain started at 0, with S=(0,infty). By symmetry pi_T(S)=1/2 for all T. So this is not an example where the chain will have a small probability for S.
> >
> > In practice (for large c and finite T) the chain is likely to spend all its time in S or none of it in S. So half the time it will approximate pi restricted to S well, and the other half of the time it will not. (In fact it is not clear how you define your approximation to pi restricted to S in the 50% of the runs of the algorithm where it does not visit S.)
> >
> > What you want the theory to stay is "given that the chain spends a substantial proportion of its time in S then it will approximate the conditional distribution of pi restricted to S well" but this is not what the result says.
> >
> > Another way of thinking about it is that there are two cases where you have poor mixing for multi-modal targets where it is difficult to move between modes. One is where you start in or near one mode (or much closer to one mode than others). In this case I can see how your result makes sense. There will be one mode you are likely to approximate well, and all other modes for which the probability assigned to that mode will be small.
> >
> > The other is where you start in the tails between two or more modes and there is randomness as to which mode you will find. In this case your result does not make sense. There will be a non-negligible probability associated with two (or more) modes. And for any run you can only approximate one of these well in practice. Thus, as with the above example, your result does not make sense.

---

> > > ### Author Response · Authors · 2023-08-18
> > > **Thank you for the response**
> > >
> > > We thank the reviewer for the response. After reading the reviewer's response, we realized that most concerns of the reviewer are due to a misunderstanding towards the definition of our distribution $\bar{\pi}_T$. We would like to first clarify this as follows.
> > >
> > > **On the distribution $\bar{\pi}_T$**:
> > >
> > > * By our definition on line 165, $\pi_t$, for $t\in[0,T]$, is the distribution of Langevin iterate $X_t$. We define $\bar{\pi}_T = \frac{1}{T} \int_0^T \pi_t dt$. Corollary 1 discusses convergence guarantees for this $\bar{\pi}_T$.
> > >
> > > * In most of the reviewer's discussions, the reviewer seems to confuse $\bar{\pi_{T}}$ with the chain distribution: choose a single initialization point $z_0$, run the algorithm for $(T+n)$ steps, select $z_{T+1}, z_{T+2},...,z_{T+n}$, and consider the empirical distribution generated from these samples.
> > >
> > > *In a word, the main difference here is that we consider the underlying distribution of LMC while the reviewer seems to consider one sample trajectory of this random process, i.e., a particular instantiation of the LMC path.*
> > >
> > > The reviewer's misunderstanding is best illustrated by considering the statement:
> > > > The other is where you start in the tails between two or more modes and there is randomness as to which mode you will find. In this case, your result does not make sense. There will be a non-negligible probability associated with two (or more) modes. And for any run, you can only approximate one of these well in practice.
> > >
> > >
> > > Contrary to what the reviewer claims, in the above example, $\bar{\pi_{T}}$ **can indeed approximate all of the modes well**. Even if a particular **instantiation of the LMC path** only visits one mode, the **underlying distribution $\bar{\pi_{T}}$** can well-approximate each mode (conditional on the support of the mode).
> > >
> > > Based on this clarification, we address other concerns of the reviewer below:
> > >
> > >
> > > > "In practice (for large c and finite T) the chain is likely to spend all its time in S or none of it in S". So half the time it will approximate pi restricted to S well, and the other half of the time it will not.
> > >
> > >  It is unclear to us what "the chain is likely to spend all its time in S" means, but we conjecture that the reviewer is talking about "if you initialize the LMC from $0$, and look at one trajectory of LMC, it will stay in S all the time (or never)", and thus this concern is due to that the reviewer confuse $\bar{\pi_{T}}$ with the chain distribution. As pointed out above, our Corollary 1 is for the averaged underlying distribution $\bar{\pi_{T}}$ of the iterates, instead of the empirical distribution formed by iterates within one trajectory. Therefore, that  "the chain spends all its time in S in one trajectory" does not indicate $\bar{\pi_{T}}(S)$ is large, and Corollary can not be applied to show that "half the time it will approximate pi restricted to S well".
> > >
> > >  > What you want the theory to stay is "given that the chain spends a substantial proportion of its time in S then it will approximate the conditional distribution of pi restricted to S well" but this is not what the result says.
> > >
> > > Neither our motivation nor our result has anything to do with "the time the chain spent in S" and thus we are confused that why this is "what we want". Still, we conjecture that the reviewer thinks that "$\bar{\pi_{T}}(S)$ is the chain distribution. If the chain spends a substantial proportion of its time in S, then $\bar{\pi_{T}}(S)$ is large and thus Corollary 1 can be applied". However, this is not true as $\bar{\pi_{T}}$ is not the chain distribution (see discussion above) and we do not intend to claim anything based on the chain distribution. What we want has been clearly stated in Corollary 1, i.e., we want the probability mass of the **underlying** distribution $\bar{\pi_{T}}$ to be small, or the conditional **underlying** distribution well approximating the conditional target distribution.
> > >
> > > Finally, we ask that the reviewer explain what he means by "does not make sense", as it is difficult to respond to a vague English statement. Does the reviewer mean
> > > 1. Corollary 1 must be wrong, or
> > > 2. Corollary 1 is vacuous?
> > >
> > > In the case of 1, we ask that the reviewer to please point to a mistake in our proof, or alternatively present a counter-example, with the density in question described clearly in mathematical notation.
> > >
> > > In the case of 2, we have already explained the significance of our result. Other reviewers have also noted that our bounds are quite relevant to understanding non-convex samples, e.g. Reviewer Edvq notes that *"In particular, the mixing rates seem to offer very good insight into the phenomenon of “metastability”, i.e. that particular modes may be well-explored while the global structure is not correct."*.

---

> > > > ### Comment · Reviewer_Lt2V · 2023-08-18
> > > >
> > > > Thanks for the clarification -- you are correct that I had misinterpreted the meaning of pi_T: and the results now appear meaningful and useful. I will adjust my score accordingly.

---

### Official Review · Reviewer_UgtB · 2023-07-04

**Soundness:** 3 good
**Presentation:** 4 excellent
**Contribution:** 3 good
**Rating:** 7
**Confidence:** 4

**Summary:**

The paper studies MCMC algorithms like the Langevin dynamics and Gibbs sampler on non-log-concave distributions. Many natural distributions are non-log-concave and multimodal, for example, mixtures of Gaussians and the posterior distribution of Gaussian mixtures. While classical results show that MCMC algorithms suffer from slow mixing on such multimodal distributions, the paper shows that when isoperimetric inequalities such as Poincare or log-Sobolev hold on a subset X of the state space, the conditional distribution of the MCMC iterate over X mixes fast to the conditional distribution of the target distribution on X. Thus, the paper shows that while MCMC algorithms converge to the true global distribution slowly, it can still converge very fast locally. For example, on a mixture of two isomorphic Gaussians, the Langevin dynamics (LMC) converges to the true conditional distribution around each mode but might put the wrong weight on the two Gaussian components. i.e. the true distribution is mu = 1/2 N(-u, sigma^2) +1/2 N (u, sigma^2) but the distribution of the LMC might be 1/3 N(-u, sigma^2) + 2/3 N(u,sigma^2). To show these results, the paper uses that for any target distribution mu, including non-log-concave ones, the LMC quickly converges to a distribution nu whose Fisher information to mu is small [Balasubramanian, Chewi, Erdogdu, Salim, Zhang—PMLR’22], then uses isoperimetric inequalities for the conditional distribution of mu on subsets of the state space to show that if the conditional distribution of mu on S satisfies isoperimetric inequality, then either nu puts small mass on S or the conditional distribution of nu on S is close to that of mu on S in Kullback-Leiber or chi-square distance (see Lemma 1, Corollary 1 for the case when the conditional distributions satisfy log-Sobolev inequalities, and Lemma 2, Theorem 2 and Corollary 2 for the case when the conditional distributions satisfy the weaker Poincare inequalities). Note that this is essentially the best statement one can hope for: for example, if mu is a mixture of two Gaussians whose centers are very far apart, and the LMC is initialized at the center of the first Gaussian, then the LMC will put almost all mass on the first Gaussian component and almost 0 mass on the second Gaussian component and one cannot have any reasonable guarantee about the conditional distribution of the LMC iterate on regions around the mode of the second component. In Theorem 2, the paper proves an analogous result for Gibbs sampler on discrete state space such as the hypercube.  For a distribution mu supported on the hypercube {0,1}^d, the Gibbs sampler operates by picking one random coordinate and flipping the value at that coordinate from 0 to 1 or 1 to 0 with probabilities chosen so that mu is the stationary distribution (see Section 5). The paper shows that if the vertices of the hypercube can be partitioned into subsets X_1,.., X_m such that on each subset, the Markov chain induced by the Gibbs sampler has a large spectral gap, then either the distribution nu produced by the Gibbs sampler puts small mass on X_i or the conditional of nu on X_ is close to that of mu on X_i.

**Strengths:**

The paper shows an interesting result using relatively simple techniques. While fast convergence of LMC iterates in Fisher information for general non-log-concave distribution and local isoperimetric inequalities are known and used in previous works (see [Balasubramanian, Chewi, Erdogdu, Salim, Zhang—PMLR’22] and [Mou, Ho, Wainwright, Bartlett, Jordan’2019]), the paper cleverly combines these two ingredients together to show local mixing of familiar MCMC algorithms likThe paper shows an interesting result using relatively simple techniques. While fast convergence of LMC iterates in Fisher information for general non-log-concave distribution and local isoperimetric inequalities are known and used in previous works (see [Balasubramanian, Chewi, Erdogdu, Salim, Zhang—PMLR’22] and [Mou, Ho, Wainwright, Bartlett, Jordan’2019]), the paper cleverly combines these two ingredients together to show local mixing of familiar MCMC algorithms like the LMC and Gibbs sampler on multimodal distributions. The main results appear to be novel and correct. I verify the proof of Lemma 1, Corollary 1, and Lemma 3.e the LMC and Gibbs sampler on multimodal distributions.

**Weaknesses:**

The result on mixtures of Gaussians requires the assumption that the Gaussian components share the same covariance. It’s unclear if this assumption is natural, and the author(s) don’t give any justification for this assumption.
The paper has a few typos. Details below.
- Proof of Lemma 1, supplement, appendix A: The claim is either $\mu(S) \leq sqrt{\epsilon}/sqrt{\alpha}$ or $Ent_{\pi|S}(\mu|S || \nu|S )\leq sqrt{\epsilon}/sqrt{\alpha}$ but the proof instead show that either $\mu(S) \leq \sqrt{\epsilon}$ or $Ent_{\pi|S}(\mu|S || \nu|S )\leq sqrt{\epsilon}/\alpha.$ The fix is simple, in line 416, it should be $\mu(S)\leq \sqrt{\epsilon}/\sqrt{\alpha}$ and in line the following displayed equation, the rhs should be $\epsilon/(\sqrt{\epsilon}/\sqrt{\alpha}) = \sqrt{\epsilon \alpha}.$
- In proof of Lemma 3, supplement, appendix C.1: c is undefined, though I believe c = min_i w_i so that c\leq w_j and cp_j \leq w_j p_j holds in the first line of the proof. The proof, Lemma 3 only proves that p |S_j satisfies LSI with constant 1/min_i w_i sigma^{-2},  but lemma 3 claims that the LSI constant is max_i w_i/min_i w_i sigma^{-2}. However, 1 \leq m max_i w_i where m is the number of components/parts in the partition of the state space, so the bound the proof achieves is only worse than the claimed bound by a factor of m, which doesn’t significantly affect the result.
-In the statement of Lemma 2, in both the main paper and supplement, pi satisfies PFI -> pi satisfies PI.
-In section 4.3.1 of the main paper, line 210, P is undefined, but I believe the author(s) mean U.
-In Lemma 5, lines 239 and 240 of the main paper, the author claims that the target distribution satisfies local LSI, but its proof (Lemma 13, line 467 of the supplement) states that the target distribution only satisfies local Poincare inequalities.



**Questions:**

-For mixtures of Gaussians, is the assumption that Gaussian components have the same covariance matrix necessary? Is this assumption natural?
-The quasi-concave condition for the Gibbs sampler to have a large spectral gap appears rather unnatural. The author(s) could consider investigating natural multimodal discrete distributions such as the Ising/Curie-Weiss model at low temperatures (see [Levin-Luczak-Peres—Probability and Related Fields’2010]).

---

> ### Author Rebuttal · Authors · 2023-08-09
>
> We thank the reviewer for the time and positive feedbacks. Your concerns are correspondingly addressed as follows.
>
>
> **Q1**: Typos &  mistakes.
>
> **A1**: We thank the reviewer for pointing out the typos. We will update our draft with the following changes:
>
> a. Lemma 1: In line 416, it should be $\mu(\mathcal{X}_i) < \sqrt{\frac{\epsilon}{\alpha}}$, and rhs of the following equation should be $\sqrt{\epsilon\alpha}$.
>
> b. Lemma 2: $\pi$ should obey $PI(\rho)$ instead of $PFI(\rho)$ in both the main paper and supplement.
>
> c. Lemma 3: In Lemma 3, $c$ should be $\min_{i} w_i$. The result should be $\pi\vert S_i$ obeys $LSI(\frac{\sigma^{-2}\min_i w_i}{m \max_i w_i})$ or just $LSI(\sigma^{-2}\min_i w_i)$, where $m$ is the number of components in the partition of the state space.
>
> d. Section 4.3.1: In line 210 and 211, $P$ should be $U$, the same as previous notation.
>
> e. Lemma 5: Here LSI is a mistake in line 239 and 240. It should be PI, which is consisitent with our proof in the supplement and with following bounds on variance instead of entropy.
>
> **Q2**: Assmuption of Gaussian components having the same covariance matrix
>
> **A2**: We thank the reviewer for pointing this out. That the components share the same covariance is necessary for our analysis. Extending the result to mixtures with different covariance matrix is possible but highly nontrivial. We mainly considered and tried two ways to tackle this more general situation:
>
> 1. Find a partition of sublevel sets that are all convex. In this way, we tried to use the same partition as we have in our paper. However, the assumption of Gaussian components having the same covariance matrix is important in our proofs, as it enables us to decompose the sample space into convex subsets.
> If we allow different covariances, the sub-level sets (i.e. $S_i = \{x: p_i(x) \geq p_j(x), \forall j \neq i\}$) can no longer be convex.  As an example, consider the following mixture of gaussian over $\mathbb{R}^2$: $p=\frac{1}{2}p_1+\frac{1}{2}p_2$, where $p_1\propto \mathcal{N}(0,\mathbb{I})$ and $p_2\propto \mathcal{N}((1,0),2\mathbb{I})$, the set $\{x: p_2(x)\ge p_1(x)\}$ can be viewed removing an ellipsoid section from the entire space, which is not convex. Therefore, Bakry-Émery criterion can no longer be applied to derive Logarithmic Sobolev Inequality. It may be still possible to derive local Logarithmic Sobolev Inequality by considering other partitions, which we leave as a future work.
>
> 2. Use identical Gaussians to approach to mixtures of K non-homogeneous Gaussians. More generally, a more complex distribution, such as a mixture of K non-homogeneous Gaussians can be approximated by a N identical Gaussians, for some N > K. The price for this reduction is increasing the number of components; but we have not worked out the optimal tradeoff for how large N needs to be.
>
> We acknowledge that sampling from more complicated mixtures is an important question for future work, and thank the reviewer for raising this point.
>
> **Q3**: Quasi-concave condition for the Gibbs sampler seems unnatural
>
> **A3**: We first explain our intuition in the quasi-convex condition. We aimed to identify a discrete analogue of log-concavity. In order to define concavity on a discrete space, we naturally resort to geodesic-concavity, in which all function super-level sets are convex with respect to geodesics (i.e. shortest paths). In our paper we show that such a condition can lower bound the spectral gap of Glauber dynamics, and hence previous analyses can readily apply.
>
> We appreciate the reviewer's refernce to closely-related and interesting work. It is possible that we generalize the CW model to multimodal setup, maybe via increasing the rank of the energy function. We think this can be an interesting direction but we won't be able to finish this within the Neurips timeline.
>
> Thank you once again for taking the time to review our paper. We sincerely hope that the response to your concerns, as well as the overall response to other reviewers’ concerns, helps assuage your concerns, and view this paper in a more favorable light.

---

> > ### Comment · Reviewer_UgtB · 2023-08-16
> > **Replying to rebuttal**
> >
> > Thank you for your detailed response.

---

### Official Review · Reviewer_7WcG · 2023-07-05

**Soundness:** 3 good
**Presentation:** 3 good
**Contribution:** 3 good
**Rating:** 6
**Confidence:** 1

**Summary:**

This work studies the convergence of MCMC algorithms for sampling from non-log-concave distributions. This is much less well-understood than the log-concave setting. The authors introduce the notion of conditional mixing, this occurs when the markov chain is close to the true (conditional) distribution when conditioned on being in some specific part of the space. They give sufficient conditions under which conditional mixing occurs. They give applications to sampling from mixture distributions of gaussians and related problems.

**Strengths:**

The authors propose a new framework to go beyond log-concave sampling, an important problem in the sampling literature and give evidence for its utility. They show that in some cases conditional mixing can appear very quickly, whereas global mixing provably takes much longer.

The paper is overall well-written.

**Weaknesses:**

I think non-expert readers could benefit from a high-level technical overview.

**Questions:**

Regarding sampling from a Gaussian mixture distribution: Could you mention where your techniques break down when allowing differing covariances for the mixture distributions?

**Limitations:**

Maybe I missed this, but I do not think limitations were mentioned.

---

> ### Author Rebuttal · Authors · 2023-08-09
>
> We thank the reviewer for the time and positive feedback. Your concerns are correspondingly addressed as follows.
>
> **Q1**: Lack of high-level technical overview.
>
> **A1**: We thank the reviewer for the suggestion. We will expand our related work to provide more overview to the problem. Due to limited space, we can not copy the entire section, but we give an outline below:
>
> a. MCMC, LMC, and their applications.
>
> b. Theoretical guarantees for LMC, and that PI / LSI / log-concavity can lead to fast mixing.
>
> c. Hardness result for LMC without log-concavity, evidenced by two mixtures of Gaussian.
>
> d. Results on non-log-concave distributions.
>
> e. discrete space MCMC.
>
> **Q2**: Difficulties for allowing different covariances for the mixture distributions.
>
> **A2**: Our current analysis is based on one important fact: the sublevel sets(i.e. $S_i = \{x: p_i(x) \geq p_j(x), \forall j \neq i\}$) are convex when the covariances are identical because the sublevel sets can be viewed as a Voronoi diagram after affine transformation. However, if we allow each mixture to have different covariance, the sub-level set may be no longer convex. As an example, consider the following mixture of Gaussian over $\mathbb{R}^2$: $p=\frac{1}{2}p_1+\frac{1}{2}p_2$, where $p_1\propto \mathcal{N}(0,\mathbb{I})$ and $p_2\propto \mathcal{N}((1,0),2\mathbb{I})$, the set $\{x: p_2(x)\ge p_1(x)\}$ can be viewed removing an ellipsoid section from the entire space, which is not convex. Therefore, Bakry-Émery criterion can no longer be applied to derive Logarithmic Sobolev Inequality. It may be, however, still possible to derive local Logarithmic Sobolev Inequality by considering other partitions. This is highly nontrivial and we leave it as a future work.
>
>
> **Q3**: No limitations were mentioned.
>
> **A3**: Thanks for pointing it out. We include limitations in the revised manuscript. For example, some limitations of this paper include:
>
> 1. lack of guarantee that our local mixing rate is sharp;
>
> 2. lack of a principled way to derive local isoperimetric inequalities beyond mixture of Gaussians with identical covariances.
>
> 3. lack of additional potential implications of fast conditional mixing.
>
> Thank you once again for taking the time to review our paper. We sincerely hope that the response to your concerns, as well as the overall response to other reviewers’ concerns, helps assuage your concerns, and view this paper in a more favorable light.

---

> > ### Comment · Reviewer_7WcG · 2023-08-14
> >
> > Thank you for your very detailed response.

---

### Official Review · Reviewer_Edvq · 2023-07-07

**Soundness:** 4 excellent
**Presentation:** 2 fair
**Contribution:** 3 good
**Rating:** 7
**Confidence:** 4

**Summary:**

This paper, following the framework of Balasubramanian et al., shows that for target distributions that are non-log-concave, isoperimetric inequalities on subsets of the state space will yield fast mixing for the conditional distributions of MCMC on that space. This adds formal justification for the observed phenomena of metastability and “local convergence”. Additionally, some analogous conditional convergence results are also reported for a random walk on a hypercube.


**Strengths:**

The definition and usage of these local isoperimetric conditions is entirely novel, as well as the local mixing results. In particular, the mixing rates seem to offer very good insight into the phenomenon of “metastability”, i.e. that particular modes may be well-explored while the global structure is not correct.

The paper is well illustrated with examples such as the Gaussian mixture and the posterior sampling example, with transparent computation of the constants. In particular, it is easy to see when the conditions of the paper hold, and to compute the resulting rate estimates.

The hypercube sampling result is novel, although it is difficult to assess this in the context of existing literature since the assumptions/results differ greatly.

Experimental evidence is also provided to quantify the phenomena of local convergence.


**Weaknesses:**

The rate of local mixing in both local LSI and local Poincare cases are quite bad. See my remark in the “Questions” section.

The discrete state space case is difficult to assess compared to other results for discrete space MCMC. If this is to be a main result in the paper, I would recommend a more detailed survey of the literature on discrete space MCMC with relevant comments be included in the Related Work.

The experiments are not surprising and seem to illustrate the same phenomena as seen in earlier works on multimodal sampling, but I would argue that they are still fairly useful illustrations in the context of this paper. In my opinion, the experiments section could be shortened.

This paper could benefit from some proofreading to catch grammatical mistakes, of which I was able to find quite a few.

To summarize, I feel that this paper makes unique contributions to the theory and intuition of MCMC algorithms, and the issues with it are relatively minor. Therefore I am happy to recommend acceptance.


**Questions:**

The local mixing seems to be at a significantly worse rate than the global mixing under e.g. LSI, which we would expect to be something like $d/\epsilon$ at least for the LSI case. On the other hand, we obtain something more like $d^2/\epsilon^4$. Is there a principled reason for this difference and could the rate potentially be improved, or is the analysis expected to be somewhat sharp.

In some prior works, e.g. [6, 23], Renyi divergence is considered as a natural “measure” of convergence, due to the analytical simplicity of the resulting expressions. Would analogous results hold for this case? Furthermore, is there any barrier to considering e.g. Latala-Oleszkiewicz or other isoperimetries in the analysis? I would imagine so given that the “Fisher information” would now need to correspond to an inequality of LO-type.

How does the discrete sampling result compare to existing MCMC results in that field? See my comment in “Weaknesses”.

Typos:
Fisher in Fisher information should always be capitalized

L. 74 space after Balasubramanian

L. 81 distribution -> distributions

L. 82. Multimode -> multimodality

L. 106 “the Langevin Monte Carlo algorithm”, function inequalities -> functional inequalities

The definition of a Lipschitz condition should be given explicitly.

Some in-line equations are a bit difficult to read and should be presented as displays, e.g. L. 231 or L. 252.

L. 286 “D” -> “Appendix D”


**Limitations:**

None beyond those raised in earlier sections.

---

> ### Author Rebuttal · Authors · 2023-08-09
>
> We thank the reviewer for taking the time to read our paper, and for the comments and suggestions. Here are our main responses:
>
> **Q1**: I would recommend a more detailed survey of the literature on discrete space MCMC with relevant comments be included in the Related Work.
>
> **A1**: Thank you for the suggestion. We will provide a more detailed literature review on discrete space MCMC in Section 2 of the revised paper.
>
> **Q2**: The experiments section could be shortened.
>
> **A2**: Thank you for the suggestion. We acknowledge that the current experiment section is a bit redundant, and will make the experiments section more condensed in the revised paper.
>
> **Q3**: The local mixing seems to be at a significantly worse rate than the global mixing under e.g. LSI. Is there a principled reason?
>
> **A3**: Thank you for the insightful question. We do not intend to claim that our rate to be sharp, and the local mixing rate may possibly be be improvable. Our main goal is to derive a mixing rate that is _polynomial over dimension_. Investigating the optimal local mixing rate is definitely an interesting future direction.
>
> **Q4**: Can the analysis be extended under Renyi divergence?
>
> **A4**: We acknowledge that Renyi divergence is a good measure of convergence, and have tried to extend our analysis under Renyi divergence but failed. In the analysis for Renyi divergence, the analogy for Fisher information $FI(\pi_t||\pi)$ is $G_{q,\pi}(\pi_t)/F_{q,\pi} (\pi_t)$. However, there is no clear relationship between $G_{q,\pi}(\pi_t)/F_{q,\pi} (\pi_t)$ and its local version due to the existence of $F_{q,\pi} (\pi_t)$ in the denominator. Therefore, even if we can show that  $G_{q,\pi}(\pi_t)/F_{q,\pi} (\pi_t)$ is small, we are unable to convert it to its local version and apply local isoperimetry.
>
> **Q5**: Is there any barrier to considering Latala-Oleszkiewicz isoperimetry?
>
> **A5**: Thank you for the suggestion. To our knowledge, existing analyses under Latala-Oleszkiewicz isoperimetry uses Renyi divergence, and the same problem in A4 will occur.
>
> Finally, thank you for pointing out the typos and minor suggestions, they have been duly revised.
>
> We sincerely hope that the response to your concerns, as well as the overall response to other reviewers’ concerns, helps assuage your concerns, and view this paper in a more favorable light.

---

> > ### Comment · Reviewer_Edvq · 2023-08-15
> > **Response**
> >
> > I thank the authors for their responses. I still have a number of concerns that I would like to be addressed.
> >
> > **A1**: I am still curious about the relevant discrete space results. Is the rate obtained here anywhere close to "optimal" in this setting? What is the expected result? Please comment if possible.
> >
> > **A3**: From a closer inspection, I see that using Proposition 1 means that $d^2/\epsilon^2$ is the best possible rate, while Lemma 1 is adding the factors of $\epsilon^2$. I would expect then that this approach is not sharp, although I don't currently have any ideas about how it could be improved. Nonetheless I hope the authors continue to investigate this question in future works.
> >
> > Regardless, I feel that this work presents some novel and genuinely useful intuition regarding the phenomena of local convergence in sampling. So long as there are improvements in the presentation of the final draft, I am happy to raise my score.

---

> > > ### Author Response · Authors · 2023-08-15
> > >
> > > We thank the reviewer for raising the score and the recognition of our contribution. We will improve the presentation and content of the final draft as promised. Below we address the additional concerns.
> > >
> > > **Q6**:  Is the rate obtained here anywhere close to "optimal" in this setting? What is the expected result?
> > >
> > > **A6**: We thank the reviewer for this question. The discrete analysis extends the continuous one, and hence we are not aware of previous works that we can directly compare against.
> > >
> > > However, a quick sanity check can be a simple random walk on graphs with radius D. This would correspond to a Gibbs sampling with uniform distribution. In this case, we know that the spectral gap is up to log factor bounded by 1/D^2, giving a similar rate as we predicted. Hence, although the assumptions may be relaxed and other dependence may be improved, the D dependence is probably tight in most setups.
> > >
> > >
> > > **Q7**: Whether the mixing rate could be improved?
> > >
> > > **A7**: We thank the reviewer for your thoughtful comment. As pointed out by the reviewer, there is an additional $\varepsilon^2$ term in Lemma 1. Such a term stems from transferring global FI to local FI, where our approach is straightforward and thus we conjecture that it can be further improved. However, we have to admit that we are still not aware of how to transfer global FI to local FI painlessly or if there is any approach to estimate local FI directly. We believe that this is an interesting question to study, and will continue to investigate this question in future works as suggested by the reviewer.

---

### Decision · Program_Chairs · 2023-09-21

**Decision:**

Accept (poster)

**Comment:**

The authors study the important problem of sampling from non-log-concave/multimodal distributions where fast mixing is generally hard to come by. They instead formulate and establish guarantees on conditional mixing, a weaker and more achievable guarantee which can be sufficient for downstream applications. The theory is well-developed and explained. The paper received high ratings, and I recommend acceptance.